# The non-coding RNA landscape of human hematopoiesis and leukemia

Adrian Schwarzer[1,2], Stephan Emmrich[3], Franziska Schmidt[3], Dominik Beck[4,5], Michelle Ng[3], Christina Reimer[3], Felix Ferdinand Adams [1], Sarah Grasedieck[6], Damian Witte[3], Sebastian Käbler[3], Jason W.H. Wong [4], Anushi Shah[4], Yizhou Huang[4], Razan Jammal[3], Aliaksandra Maroz[3], Mojca Jongen-Lavrencic[7], Axel Schambach[1,8], Florian Kuchenbauer[6], John E. Pimanda [4,5], Dirk Reinhardt[9], Dirk Heckl[3] & Jan-Henning Klusmann [3]

Non-coding RNAs have emerged as crucial regulators of gene expression and cell fate decisions. However, their expression patterns and regulatory functions during normal and malignant human hematopoiesis are incompletely understood. Here we present a comprehensive resource defining the non-coding RNA landscape of the human hematopoietic system. Based on highly specific non-coding RNA expression portraits per blood cell population, we identify unique fingerprint non-coding RNAs—such as LINC00173 in granulocytes—and assign these to critical regulatory circuits involved in blood homeostasis. Following the incorporation of acute myeloid leukemia samples into the landscape, we further uncover prognostically relevant non-coding RNA stem cell signatures shared between acute myeloid leukemia blasts and healthy hematopoietic stem cells. Our findings highlight the importance of the non-coding transcriptome in the formation and maintenance of the human blood hierarchy.

[1] Institute of Experimental Hematology, Hannover Medical School, Hannover 30625, Germany. [2] Clinic for Hematology, Oncology, Hemostaseology and Stem Cell Transplantation, Hannover Medical School, Hannover 30625, Germany. [3] Pediatric Hematology and Oncology, Hannover Medical School, Hannover 30625, Germany. [4] Lowy Cancer Research Centre and the Prince of Wales Clinical School, UNSW Australia, Sydney, NSW 2052, Australia. [5] Centre for Health Technologies and the School of Software, University of Technology, Sydney, NSW 2007, Australia. [6] Department of Internal Medicine III, University Hospital of Ulm, Ulm 89081, Germany. [7] Department of Hematology, Erasmus University Medical Center, Rotterdam 3015, The Netherlands. [8] REBIRTH Cluster of Excellence, Hannover Medical School, Hannover 30625, Germany. [9] Clinic for Pediatrics III, University Hospital Essen, Essen 45122, Germany. Adrian Schwarzer and Stephan Emmrich contributed equally to this work. Correspondence and requests for materials should be addressed to J.-H.K. (email: klusmann.jan-henning@mh-hannover.de)

To maintain hematopoietic stem cell (HSC) homeostasis and lifelong blood production, a complex interplay of growth factors, signaling cascades, and transcription factors controls the balance between self-renewal, proliferation, quiescence, and differentiation. Deregulation of this critical balance results in myelodysplasia, myeloproliferation, or leukemia. Despite the discovery of an enormous number and diversity of transcripts from the previously ignored non-protein-coding genome[1], our knowledge remains limited regarding how non-coding RNAs (ncRNAs) are involved in this interplay. In particular, ncRNAs encompass a plethora of small regulatory RNAs including microRNAs (miRNAs), as well as tens of thousands of polyadenylated and non-polyadenylated long ncRNAs (lncRNAs)[1]. LncRNAs can be antisense, intronic, intergenic, and overlapping with respect to protein-coding loci, and can affect multiple stages of gene regulation including chromatin modification, chromatin structure, and mRNA and protein biogenesis during differentiation and development[1, 2]. Consistent with this model, lncRNA expression is tightly controlled and exhibits even higher cell specificity than proteins—including lineage-determining transcription factors[3, 4].

While miRNAs are established regulators of hematopoiesis and leukemogenesis[5, 6], lncRNAs as a class of transcripts remain largely undescribed. Even in the case of known lncRNAs,

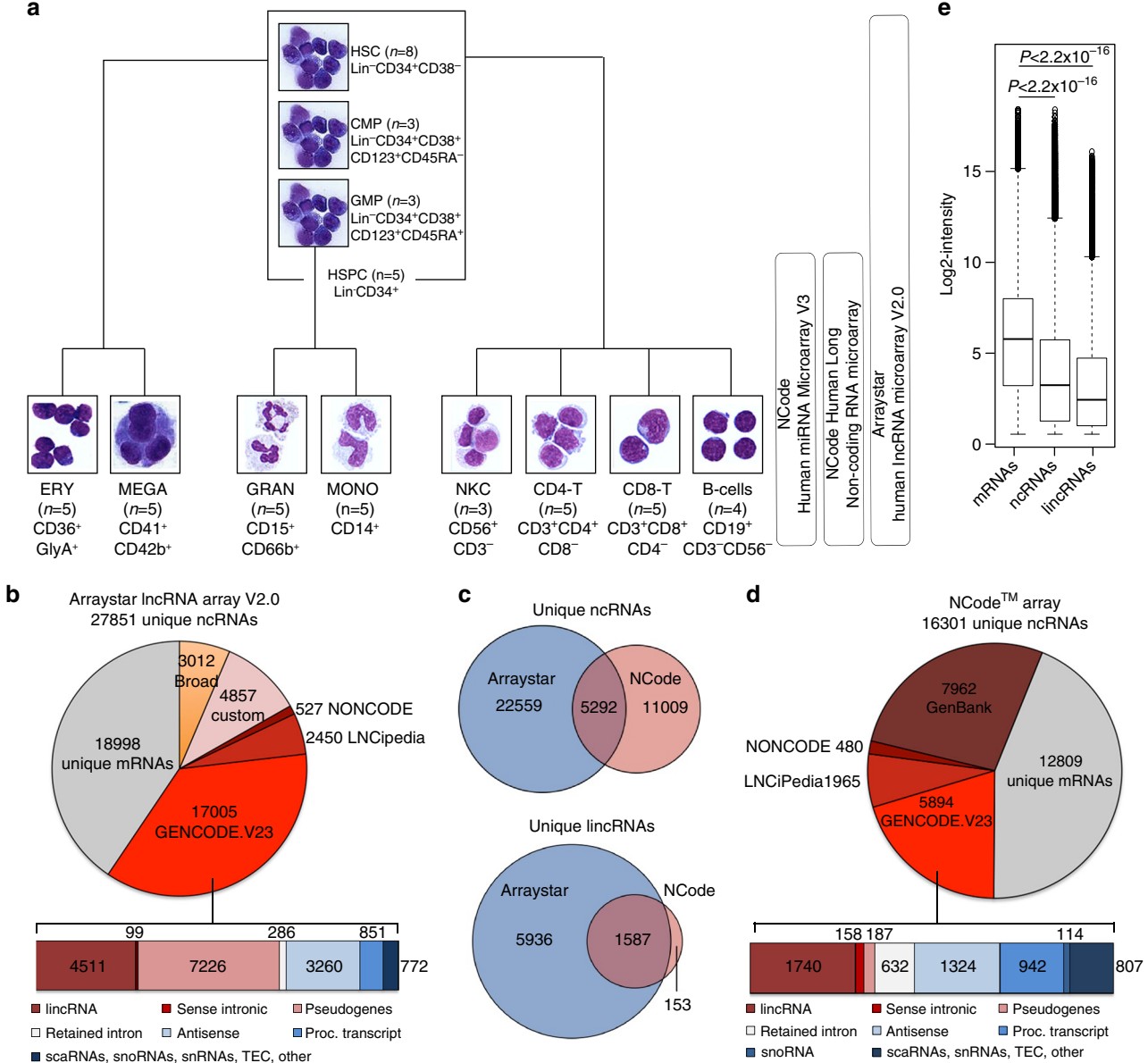

**Fig. 1** Microarray-based profiling of the ncRNA landscape in normal hematopoiesis. **a** Blood cell populations purified by multicolor flow cytometry from different healthy individuals: hematopoietic stem cells (*HSCs*), common myeloid progenitor cells (*CMPs*), granulocyte-monocyte progenitor cells (*GMPs*), megakaryocytes (*MEGA*), erythroid precursors (*ERY*), granulocytes (*GRAN*), monocytes (*MONO*), CD4 and CD8 T-cells, NK cells (*NKC*), and B-cells. Cytospins were prepared after sorting. For HSCs, CMPs, and GMPs representative cytospins from in vitro expanded CB HSPCs are depicted. **b** Annotation, distribution, and functional classes of the Arraystar Human lncRNA Microarray V2.0 probes according to the indicated databases. **c** Overlapping features between the Arraystar Human lncRNA Microarray V2.0 and the NCode Human Long Non-coding RNA microarray. **d** Annotation, distribution, and functional classes of the NCode Human Long Non-coding RNA microarray probes. **e** Box plots of log2-probe intensities for mRNAs (*n* = 978,840), ncRNAs (*n* = 784,530), and lincRNAs (*n* = 90,855) from all NCode Human Long Non-coding RNA microarrays. *P*-values were calculated using the two-tailed Welsh's *t*-test

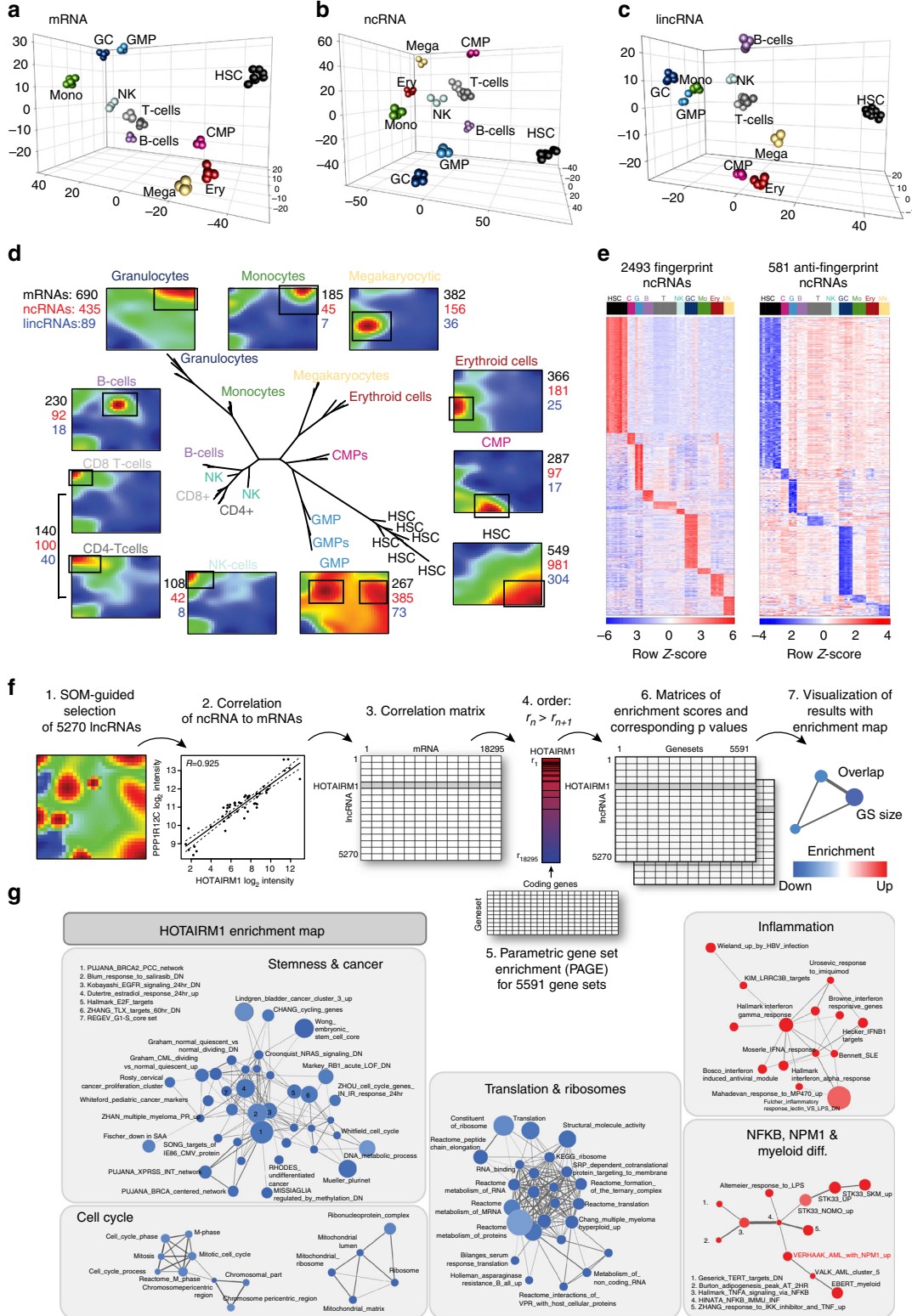

**Fig. 2** Distinct ncRNA expression profiles characterize cells of the different human blood lineages. All depicted data refer to the Arraystar Human lncRNA Microarray V2.0 platform. **a–c** *t*-SNE of all samples using the most variable **a** mRNAs (3926), **b** ncRNAs (3151), and **c** lincRNAs (767). **d** Self-organizing maps (*SOMs*) trained using the 17,655 most variable mRNAs and ncRNAs in 11 sample groups. *Black rectangles*: group-specific overexpression spots. *Center*: neighbor-joining tree built using the 68 lineage-specific spot metagenes. **e** Heatmaps of (*left*) 2493 fingerprint ncRNAs and (*right*) 581 anti-fingerprint ncRNAs, defined by integrating SOM and limma analyses. **f** Guilt-by-association workflow for the fingerprint/anti-fingerprint ncRNAs and all protein-coding genes. **g** Enrichment map network analysis for *HOTAIRM1* (FDR < 0.05, see the methods section for details). Circle size corresponds to the size of the gene set, and connecting line thickness represents the degree of similarity between two gene sets. *Red* and *blue nodes* indicate positive and negative correlation to *HOTAIRM1* expression, respectively. Gene set labels printed in *bold* indicate a similar association (FDR < 0.05) observed in at least one AML validation cohort

little functional information exists about their contribution to hematopoietic processes and malignant transformation, with the exception of a handful of well-characterized examples. Several of these examples have been shown to control the maintenance of mouse long-term HSCs[7, 8] or the emergence of hematologic cancers[9–11].

Given the poor cross-species conservation and species-specific expression patterns of ncRNAs, it is crucial to study their regulation and regulatory function in humans, in order to develop new strategies for leukemia treatment and regenerative medicine. However, a systematic profiling and functional investigation of known ncRNAs in the human hematopoietic system—including a comparison to malignant leukemic blasts—has yet to be reported. Here, we present a gene expression-based landscape of the normal human hematopoietic hierarchy, generated by using short and long ncRNAs, mRNAs from purified HSCs, and their differentiated progenies. With this resource, we identify fingerprint ncRNAs for each blood lineage and predict their functions during blood formation. By mapping pediatric AML patient samples onto this landscape, we identify a stem cell signature of ncRNAs upregulated in HSCs and AML blasts, as well as AML subtype-specific ncRNAs. Our data and bioinformatic analyses constitute a publically available resource for exploring the transcriptional networks that underlie normal and malignant hematopoiesis.

## Results

**Establishing a ncRNA expression atlas of human hematopoiesis.** In order to establish a human ncRNA hematopoietic expression atlas, we profiled the expression of known ncRNAs in 12 distinct blood cell populations purified by multicolor flow cytometry from three to eight healthy donors. For each blood cell population, RNA was hybridized onto three microarray platforms (Fig. 1a), yielding a coverage of 38,860 unique ncRNAs, 20,466 mRNAs, and 900 miRNAs on 146 arrays (Fig. 1b–d). All array probes were re-annotated using GENCODE v23, LNCipedia 3.1, NONCODEv4, the human lincRNA body map[4], and GenBank, in that order of priority. The quantified transcripts were comprised of mRNAs and different types of ncRNAs including long-intervening ncRNAs (lincRNAs), pseudogenes, antisense transcripts, retained introns, and small nucleolar RNAs (snoRNAs) (Fig. 1b–d). As previously shown[4], the mean expression level of ncRNAs was nearly twofold lower ($P_{Welsh's} < 2.2 \times 10^{-16}$, Fig. 1e) compared to mRNAs. Principle component analysis (PCA) demonstrated high inter-platform concordance between the different array platforms (Supplementary Fig. 1a–c). Furthermore, the identity of the input populations and the performance of the microarray platforms were confirmed by correlating our mRNA expression profiles with those previously published in the differentiation map of (human) hematopoiesis (DMAP)[12] (Supplementary Fig. 1d–g). Thus, we confirmed the quality of our expression data and input cell populations, allowing for subsequent systematic analysis of ncRNAs in the hematopoietic hierarchy.

**Unique ncRNA expression profiles characterize blood lineages.** To test whether the blood cell populations could be distinguished based on mRNA, ncRNA, or lincRNA expression, we performed non-linear dimensionality reduction by $t$-distributed stochastic neighbor embedding ($t$-SNE)[13]. The $t$-SNE representation of the data showed robust separation of the samples according to cell type, not only implying specific mRNA signatures (Fig. 2a and Supplementary Fig. 2a), but also unique ncRNA (Fig. 2b and Supplementary Fig. 2b) and lincRNA (Fig. 2c and Supplementary Fig. 2c) expression profiles for each population. We further structured the data set using self-organized maps (SOMs), which

combine sample- and gene-centered analyses[14]. In a SOM every gene is plotted onto a 2D grid in such a way that genes with similar expression profiles map to the same region of the graph, forming spots of co-expressed genes. With this method, we obtained unique expression portraits for each blood cell type, from which coordinately upregulated mRNAs, ncRNAs, and lincRNAs could be extracted from hotspots in the portraits (Fig. 2d and Supplementary Fig. 2d). Furthermore, neighbor-joining on the unique SOM-expression profiles constructed a tree of the samples that recapitulated the hematopoietic tree (Fig. 2d and Supplementary Fig. 2d). Constructing the tree based only on ncRNA expression produced similar results (Supplementary Fig. 2e).

To obtain high confidence lineage-specific ncRNA and lincRNA signatures for each blood cell type, we determined the overlap between SOM analyses and empirical Bayes methods (linear models for microarray analysis (limma))[15]. This overlap contained a total of 2493 fingerprint and 581 anti-fingerprint ncRNAs (Fig. 2e and Supplementary Fig. 2f, g, Supplementary Data 1, 2). The cell type specificity of the top-ranked HSC fingerprint lincRNAs was validated by qRT-PCR (Supplementary Fig. 2h). Overall, the highly cell-type-specific ncRNA expression we observe in the human hematopoietic system implies the tight regulation and coordinated function of this class of RNAs.

**"Guilt-by-association" approach predicts ncRNA functions.** Aiming to infer putative functions for lineage-associated ncRNAs during differentiation, we constructed a correlation matrix between the expression profiles of the fingerprint/anti-fingerprint ncRNAs and 18,295 protein-coding genes (Fig. 2f). We hypothesized that ncRNAs and coding genes belonging to the same biological pathways are likely coordinately regulated. In a guilt-by-association approach[16], the correlation data were aggregated by parametric analysis of gene set enrichment (PAGE)[17] to compute the associations of each ncRNA with over 6000 gene sets[18] (Supplementary Data 3). This yielded more than 70,000 significant ncRNA-gene set interactions (false discovery rate (FDR) < 0.01), which could be further interrogated by clustering functional modules (Fig. 2f). For *HOTAIRM1*, a well-known granulocyte fingerprint lncRNA[19], the analysis predicted association with inflammatory and innate immune response pathways, and showed a strong correlation with gene sets upregulated in *NPM1*-mutated AML (Fig. 2g). Furthermore, the algorithm predicted a negative association between *HOTAIRM1* and ribosome biogenesis, pluripotency and cell cycle progression, which is consistent with *HOTAIRM1* being a negative cell cycle regulator during myeloid differentiation[20].

We validated our approach in two independent data sets of more than 600 AML samples[21, 22], demonstrating remarkable stability with an overlap of 80% of all associated gene sets (Supplementary Fig. 3a, b, Supplementary Data 4). Most importantly, as predicted by our data set, AMLs with *NPM1* mutations were characterized by significantly higher expression of *HOTAIRM1* compared to *NPM1*-wild type samples in both AML cohorts ($P_{Welsh's} < 10^{-7}$ and $P_{Welsh's} < 10^{-15}$; Supplementary Fig. 3c, d). In summary, for every blood cell population we were able to identify fingerprint ncRNAs, to which we assigned potential functions using a guilt-by-association approach coupled with gene set enrichment analyses (GSEA). The capacity of this pipeline to infer putative functions and generate testable hypotheses was exemplified with the well-studied lncRNA *HOTAIRM1*.

**Prediction of novel ncRNA regulators of granulopoiesis.** To test whether our expression resource and established bioinformatic pipeline can identify novel functionally important

fingerprint lncRNAs and guide hypothesis-driven research, we focused on the granulocyte signature ncRNAs. Within this signature, we identified a set of lncRNAs—including *HOTAIRM1*—that are absent from HSCs but become gradually upregulated from CMPs to GMPs to granulocytes (Fig. 3a).

To maximize coverage of the non-coding transcriptome and to confirm that the use of microarray platforms did not bias our

analyses of myelopoiesis, we performed RNA-sequencing (RNA-seq) in myeloblasts, promyelocytes, metamyelocytes, and mature neutrophils to represent the myeloid differentiation path[23] (Fig. 3b, c). Whereas RNA-seq performed equally well as arrays for the detection of coding genes, we found that low read counts impaired the ability of RNA-seq to reliably estimate the abundance of many ncRNAs. The combination of two array

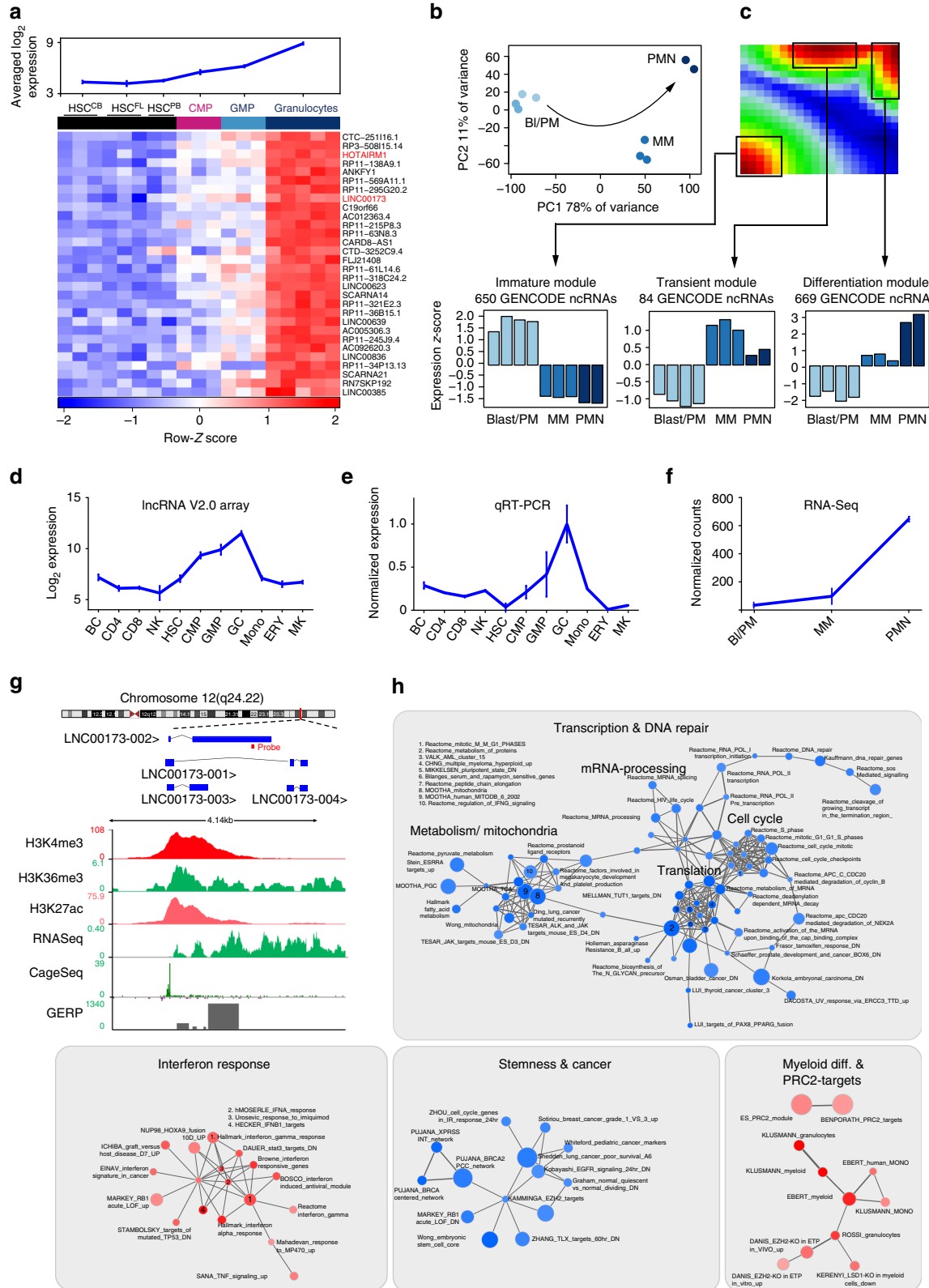

platforms yielded more than a twofold higher coverage of GENCODE-annotated ncRNAs (18,280) or lincRNAs (4228) than RNA-seq (7759 ncRNAs and 1502 lincRNAs; Supplementary Fig. 4a). Additional 2569 GENCODE-annotated ncRNAs were detected by RNA-seq, but were not captured by the arrays. To extract modules of co-regulated ncRNAs in the RNA-seq data set, we again trained a SOM. This led to the identification of three robust co-expression modules of ncRNAs upregulated early, transiently, or late during myeloid differentiation (Fig. 3c, Supplementary Fig. 4b–d, and Supplementary Data 5).

We reasoned that ncRNAs which are gradually upregulated from HSCs to CMPs to GMPs to granulocytes (microarray platforms) and from myeloblasts, promyelocytes, metamyelocytes, and mature neutrophils (RNA-seq) may be early regulators of granulopoiesis. Of these, LINC00173 was the lincRNA with the most specific expression in mature granulocytes (Fig. 3a, d–f). LINC00173 is encoded on the long arm of chromosome 12 and exists in four major isoforms (Fig. 3g). In human neutrophils, the LINC00173 locus shows promoter- (H3K4me3, H3K27ac) and elongation-associated (H3K36me3) histone modifications[24] as well as a strong cap analysis of gene expression (CAGE) signal at its transcriptional start site[25] (TSS) indicative of active transcription (Fig. 3g) Whole-genome bisulfite sequencing data further showed demethylation of the LINC00173 locus in differentiated blood cells. Meanwhile, the absence of CAGE-seq signals, H3K4me3, H3K27ac, or H3K36me3 in B-cells, T-cells, erythroblasts, and monocytes (Supplementary Fig. 5a) validate LINC00173 as an actively transcribed and regulated gene specifically in granulocytes. CAGE-seq data from more than 700 human tissue samples and cell lines[25] confirmed that LINC00173 has more than a tenfold higher expression in granulocytes than in any other human tissue (Supplementary Fig. 5b). Notably, LINC00173 shows substantial sequence conservation both at the TSS ($P < 10^{-20}$; RS score > median background[25]) and in the gene body ($P < 10^{-18}$; RS score > median background[25]) based on its overlap with Genomic Evolutionary Rate Profiling (GERP)-elements[26] (Fig. 3g).

Our guilt-by-association approach revealed LINC00173 to be co-expressed with genes involved in interferon response, myeloid differentiation, and polycomb repressive complex 2 (PRC2)-regulated networks (Fig. 3h). Inversely, LINC00173 expression was negatively associated with gene sets involved in stemness and cancer, as well as cell cycle progression (Fig. 3h). Together, the guilt-by-association network and the expression pattern of LINC00173 suggest the involvement of this lincRNA in regulating the proliferation and maturation of granulocytes.

**LINC00173 controls myeloid differentiation**. The repression of stemness and proliferation programs is an essential requirement for granulocytic differentiation, and has been linked to chromatin remodeling[27, 28]. The positive correlation between LINC00173 and PRC2-regulated networks and the negative correlation with stem cell and cell cycle networks identified by our bioinformatic

pipeline led us to hypothesize that LINC00173 represses the latter expression programs during granulopoiesis. To test this hypothesis, we knocked down LINC00173 in human CD34$^+$ hematopoietic stem and progenitor cells (HSPCs) using two different validated shRNAs (Supplementary Fig. 5c). This caused a defect in granulocytic differentiation in vitro, as indicated by a significantly reduced proportion of CD66b$^+$ granulocytic cells compared to the non-targeting shRNA-transduced control cells (Fig. 4a). Morphologic analyses and leukocyte peroxidase (POX) staining confirmed an increase of myeloid precursors and a concomitant decrease of mature POX$^+$ granulocytes with multi-lobed nuclei (Fig. 4a). Functionally, LINC00173-knockdown cells showed significantly decreased phagocytic capacity—indicating impaired functionality as is expected with perturbed maturation (Fig. 4b and Supplementary Fig. 5d). However, while we observed reduced proliferation and myeloid colony formation, erythroid colony formation was almost unaffected by LINC00173 knockdown (Fig. 4c, d). This not only suggests a role for LINC00173 during proper granulocytic differentiation, but also that LINC00173 is already required for the growth and maintenance of early myeloid progenitors or precursors.

Next, we applied CRISPR-interference (CRISPRi)[29] for transcriptional repression of LINC00173 in the NB4 leukemia cell line. NB4 cells have an intrinsic block of granulocytic differentiation at the promyelocyte stage. Repression of LINC00173 using two different sgRNAs (Supplementary Fig. 5e) reduced proliferation in six independent NB4:dCas9-KRAB monoclones (Fig. 4e), underlining the importance of LINC00173 at an early stage of myelopoiesis.

Localization studies can provide first insights into the molecular functions of lncRNAs. For LINC00173 we therefore performed RNA fractionation followed by qRT-PCR and RNA fluorescence in situ hybridization. These experiments revealed the localization of LINC00173 in the nucleus, similar to the X-inactivating XIST and tumor suppressor MALAT1 lncRNAs (Fig. 4f, g and Supplementary Fig. 5f, g).

To capture early transcriptional changes mediated by LINC00173, we examined the effects of LINC00173 knockdown on gene expression profiles in transduced CD34$^+$ HSPCs. GSEA (Fig. 4h) and a comparison of the leading-edge genes with the human DMAP data set[12] (Fig. 4i) revealed an upregulation of gene sets related to stemness, megakaryopoiesis, and erythropoiesis upon LINC00173 knockdown. These data indicate that the negative association between LINC00173 and stemness gene sets observed in the guilt-by-association approach is indeed the consequence and not the cause of LINC00173 expression.

Since our guilt-by-association approach also revealed the co-expression of LINC00173 with genes involved in PRC2-associated networks (Fig. 3h), we speculated that LINC00173 associates with components of PRC2, as has been shown for several other lncRNAs[30]. Indeed, RNA immunoprecipitation (RIP) followed by qRT-PCR using two independent antibodies in two different cell lines indicated binding between LINC00173 and the EZH2 subunit of PRC2 (Fig. 4j and Supplementary Fig. 5h).

**Fig. 3** LINC00173 is a granulocyte-specific lincRNA. **a** Averaged expression (*top*) and heatmap of granulocyte fingerprint ncRNAs (top 30 without pseudogenes) which show increasing expression from HSCs to CMPs to GMPs. **b** RNA-seq of human myelopoiesis: PCA on the 1373 most variable ncRNAs in the data set. The arrow indicates the main trajectory of myeloid maturation. *Bl/PM* blasts/promyelocytes, *MM* metamyelocytes, *PMN* polymorphonuclear neutrophils. **c** SOM representation of RNA-seq data set revealing three spots of co-regulated metagenes (modules), whose expression properties are depicted in the bar charts below. **d–f** LINC00173 expression normalized to granulocytes as measured by **d** the Arraystar Human lncRNA Microarray V2.0 ($n = 3–5$ per data point), **e** qRT-PCR ($n = 3$), and **f** RNA-Seq ($n = 2–4$). Error bars indicate ± s.e.m. **g** The LINC00173 gene locus depicting the array probe and alternative isoforms (according to ENSEMBL GRCh38.p5), together with UCSC genome browser tracks (http://genome.ucsc.edu; *assembly*: GRCh38/hg38) of RNA-Seq and ChIP-seq data (BLUEPRINT)[24], CAGE-Seq Signals (FANTOM5)[25], and sequence conservation (GERP-elements)[26] in mature human neutrophils. **h** Guilt-by-association results for LINC00173. Circle size corresponds to the size of the gene set, and connecting line thickness represents the degree of similarity between two gene sets. *Red* and *blue* nodes indicate positive and negative correlation to LINC00173 expression, respectively

PRC2 is required for proper lineage specification during hematopoietic differentiation, and acts by silencing a "legacy" of stem cell genes and suppressing alternative cell fates via trimethylation of H3K27[27]. ChIP-seq demonstrated that *LINC00173* knockdown in CD34[+] HSPCs results in differential H3K27 trimethylation at the promoter regions of leading-edge genes (Fig. 4i) related to stemness, megakaryopoiesis, and erythropoiesis during early myeloid specification (Fig. 4k). Among these was the *HOXA7-HOXA10* locus containing *HOXA7*, *HOXA9*, and *HOXA10*—homeodomain-containing transcription factors with important roles in the expansion of HSCs and AML blasts[31] (Fig. 4k). The promoter of the Rho GTPase-activating protein SYDE1 also showed loss of H3K27me3 (Fig. 4k).

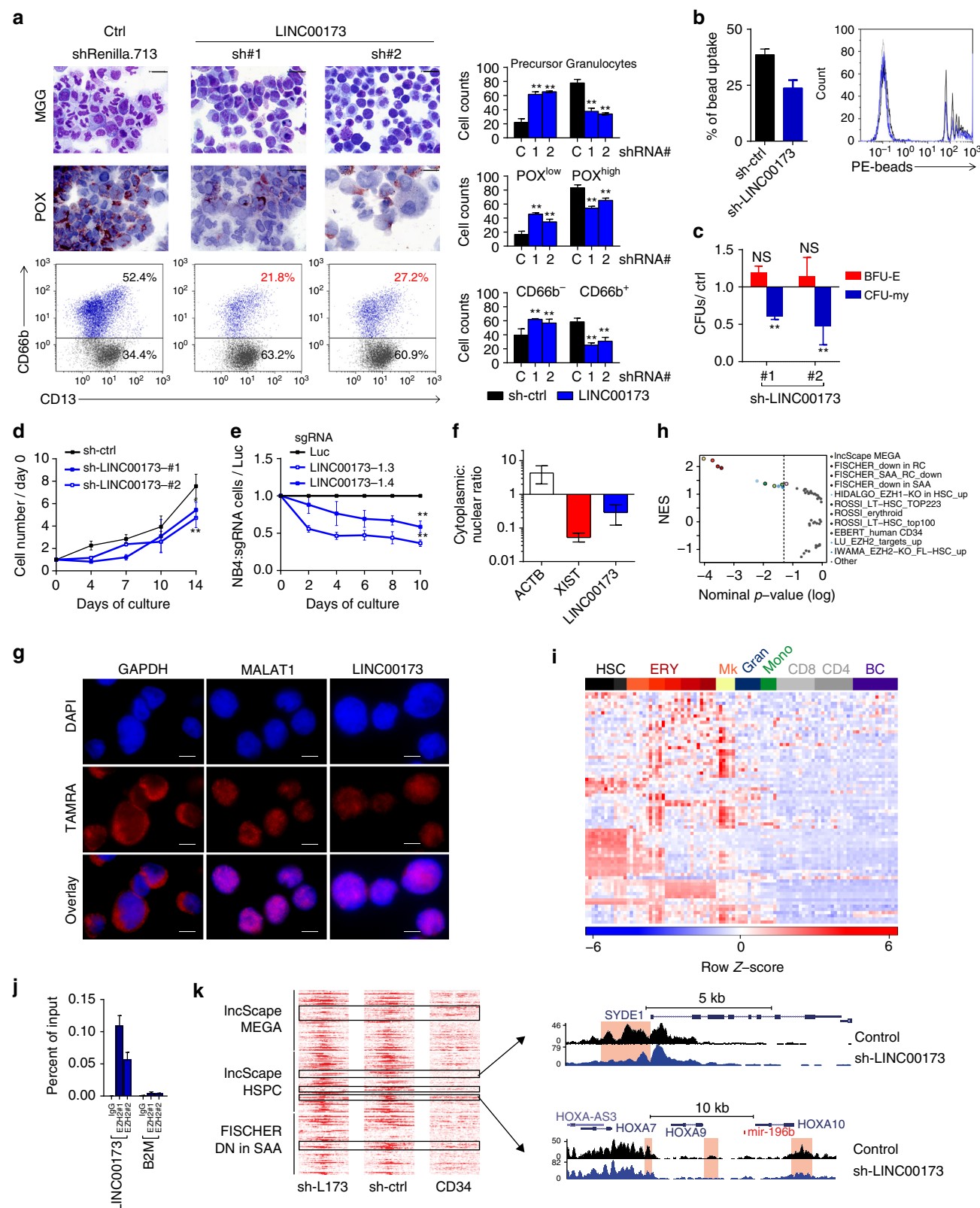

In total, our data strongly suggests the contribution of *LINC00173* to lineage fidelity in complex with PRC2. We have shown this link through a wide range of cellular and molecular techniques, culminating in proof of a direct interaction between *LINC00173* and EZH2, and changes in H3K27 at *HOX* promoters upon *LINC00173* knockdown.

**DLK1-DIO3 cluster ncRNAs are upregulated in megakaryopoiesis.** As previously shown, each blood cell population possesses a distinct miRNA expression profile[5], which we confirmed with our data set using *t*-SNE, limma, and SOM-expression portraits (Fig. 5a, b, Supplementary Fig. 6a and Supplementary Data 6). Next, we integrated hematopoietic miRNA and ncRNA expression data from our resource and coupled them to chromosomal positioning, in order to detect coordinated expression changes of genes located within specific cytogenetic bands. This multi-dimensional approach uncovered a strong and highly coordinated upregulation of several ncRNA classes in the *DLK1-DIO3* locus on human chromosome 14, specifically in megakaryocytes (average $\log_2$-fold change > 5, average adj. $P < 10^{-8}$; moderated *t* test[15], confirmed by qRT-PCR; Fig. 5c, d and Supplementary Fig. 6b–d). The *DLK1-DIO3* locus (Fig. 5e) is imprinted and harbors three paternally expressed coding genes (*DLK1*, *RTL1*, and *DIO3*), as well as numerous maternally expressed ncRNAs—all of which are regulated by a common *cis*-element (IG-DMR) and transcribed as a single huge polycistronic transcript[32, 33]. Among the ncRNAs are several lncRNAs, a box C/D snoRNA cluster, and 54 miRNAs, including the *miR-127~136* cluster (7 miRNA-members), the *miR-379~410* megacluster (42 miRNAs), and intronic miR-770. While the lncRNAs of the homologous mouse *Dlk1-Gtl2* (alias *Dlk1-Meg3*) locus are specifically expressed in murine LT-HSCs[7], we discovered that this specificity is not conserved in the human system. Instead several of these ncRNAs and their isoforms are upregulated in megakaryocytes compared to HSCs (Supplementary Fig. 6c, d; validated by qRT-PCR in Fig. 5c, d) and are retained in thrombocytes (Fig. 5c). In CD41[+] megakaryocytes the locus is marked by H3K4 trimethylation, as well as H3K27 acetylation at the promoter region and H3K36 trimethylation along the length of the ncRNA cluster[24], confirming active transcription in these cells (Fig. 5e).

To investigate whether the miRNAs and lncRNAs of the human *DLK1-DIO3* locus play a role in the differentiation and maintenance of megakaryocytes, we performed gain- and loss-of-function studies using human CD34[+] HSPCs. We ectopically expressed four miRNAs representing different clusters and/or miRNA-families: miR-770 located in the intron of *MEG3*, miR-136 from the miR-127~136 cluster, and miR-379 and miR-410 as

the first and last miRNAs from the megacluster (Supplementary Fig. 7a). During megakaryocytic differentiation we noted a sharp increase of CD41[+]/CD42b[+] megakaryocytic cells upon forced expression of each of the four miRNAs (Fig. 5f); meanwhile, their effects on proliferation were subtle (Supplementary Fig. 7b). The impact of the *DLK1-DIO3* locus on megakaryopoiesis was reinforced by loss-of-function experiments, with shRNAs directed against the *MEG3*, *MEG8*, *MEG9* lncRNAs, and the coding gene *DLK1* (two shRNAs per gene; Supplementary Fig. 7c). Knockdown of each of the four genes interfered with megakaryocytic differentiation, as evidenced by a reduction of differentiated CD41[+]/CD42b[+] megakaryocytes in culture, with little effect on proliferation (Fig. 5g and Supplementary Fig. 7d). Similarly, a reduction of CD41[+] CFU-megakaryocytes was observed upon knockdown of the three lncRNAs (Supplementary Fig. 7e, f). In conjunction with our integrated microarray-based miRNA/ncRNA expression analysis, these gain- and loss-of-function experiments define a novel but important function for the *DLK1-DIO3* locus during differentiation along the megakaryocytic lineage—thereby classifying the locus as a positive regulator of megakaryopoiesis.

**Self-organizing maps reveal ncRNA stem cell signatures in AML.** To extend our findings and our resource to malignant hematopoiesis, we incorporated 46 pediatric AML samples into the human hematopoietic ncRNA expression atlas. The AML samples included Down syndrome (DS) and non-DS acute megakaryoblastic leukemia (AMKL; i.e., AML FAB M7), core-binding factor (CBF) AMLs (inv[16] and t[8;21]) and *MLL*-rearranged AMLs (t[9;11] and t[10;11]). The identities of the underlying AML subgroups in the corresponding samples were confirmed by single-sample pathway activity analysis[34] on coding genes (Supplementary Fig. 8a), enabling a credible systematic analysis of the ncRNAs associated with each subgroup. Indeed, we identified ncRNA signatures specific to DS-AMKL, AMKL, t(8;21), inv(16), and *MLL*-rearranged samples (Fig. 6a).

Next, we used *t*-SNE to map the pediatric AML samples onto the ncRNA expression landscape of normal hematopoietic differentiation. Density-based clustering (densVM)—a machine-aided subset identification algorithm[35]—correctly identified all groups of normal blood cells, which served as landmarks on the two-dimensional (2D) *t*-SNE landscape (Fig. 6b and Supplementary Fig. 8b, c). The AML samples were divided into four distinct groups on this landscape, based on their ncRNA expression (Fig. 6b and Supplementary Fig. 8b, c). Two groups mapped closely with healthy HSCs, which we termed stem cell AML (SC-AML) groups I and II. The SC-AML group I contained mainly DS- and non-DS-AMKL samples (six out of eight samples),

**Fig. 4** *LINC00173* is a novel regulator of granulocytic development. **a–d** RNAi (shRNA)-mediated knockdown of *LINC00173* in CD34[+] HSPCs in vitro. **a** Granulocytic in vitro differentiation (day 14). Upper panel: May-Grünwald Giemsa (*MGG*) staining; scale bars 20 μm. Middle panel: neutrophil peroxidase (*POX*) staining; scale bar 20 μm. Lower panel: flow cytometric analysis of CD66b and CD13 surface marker expression. The bar graphs (right) show the mean ± s.d. of three independent experiments. **b** Percentage of bead-positive cells in a phagocytosis assay. The histogram depicts the fluorescence intensity. **c** Number of BFU-E and CFU-G/M (CFU-my) in methylcellulose-based CFU-assays normalized to the non-silencing shRNA control (*ctrl*). **d** Number of shRNA-transduced cells during granulocytic in vitro differentiation normalized to day 0. **e** Ratio of RFP657[+] sgRNA-transduced vs. untransduced cells relative to the non-targeting control (sgRNAs against luciferase), using monoclonal NB4 cell lines stably expressing dCas-KRAB (*n* = 6). **f** Cytoplasmic to nuclear ratio of *LINC00173* determined by qRT-PCR on fractionated RNA from THP-1 cells. **g** RNA FISH with tiled biotinylated probes in THP-1 cells; scale bars 10 μm. **h** GSEA results for 52 hematopoiesis-associated gene sets upon *LINC00173* knockdown in CD34[+] HSPCs. The plot shows normalized enrichment scores (*NES*) against nominal *P*-values of the normalized enrichment score[18], dotted line: *P* = 0.05. **i** Heatmap showing expression of leading-edge genes from the "Fischer_DOWN IN SEVERE APLASTIC ANEMIA" gene set across the human DMAP data set[12]. **j** RIP in NB4 cells using two different antibodies, followed by qRT-PCR to detect binding of EZH2 to *LINC00173*. Data are presented as percent of input in comparison to *B2M*. **k** ChIP-seq density heatmaps for H3K27me3 in promoter regions of leading-edge genes from the indicated gene sets upon *LINC00173* knockdown. shRNA-transduced CD34[+] HSPCs (sh-L173 and sh-CTRL) are compared to untransduced and uncultured CD34[+] HSPCs. Clusters of promoters with differential H3K27me3 marks are highlighted. Representative stem cell-specific genes are shown (right). Data are presented as mean ± s.d. **a–f**, **j**, or s.e.m. **c**, **e**. *P < 0.05; **P < 0.01; *ns* not significant; *P*-values were calculated using one-way ANOVA with Dunnett's post hoc test

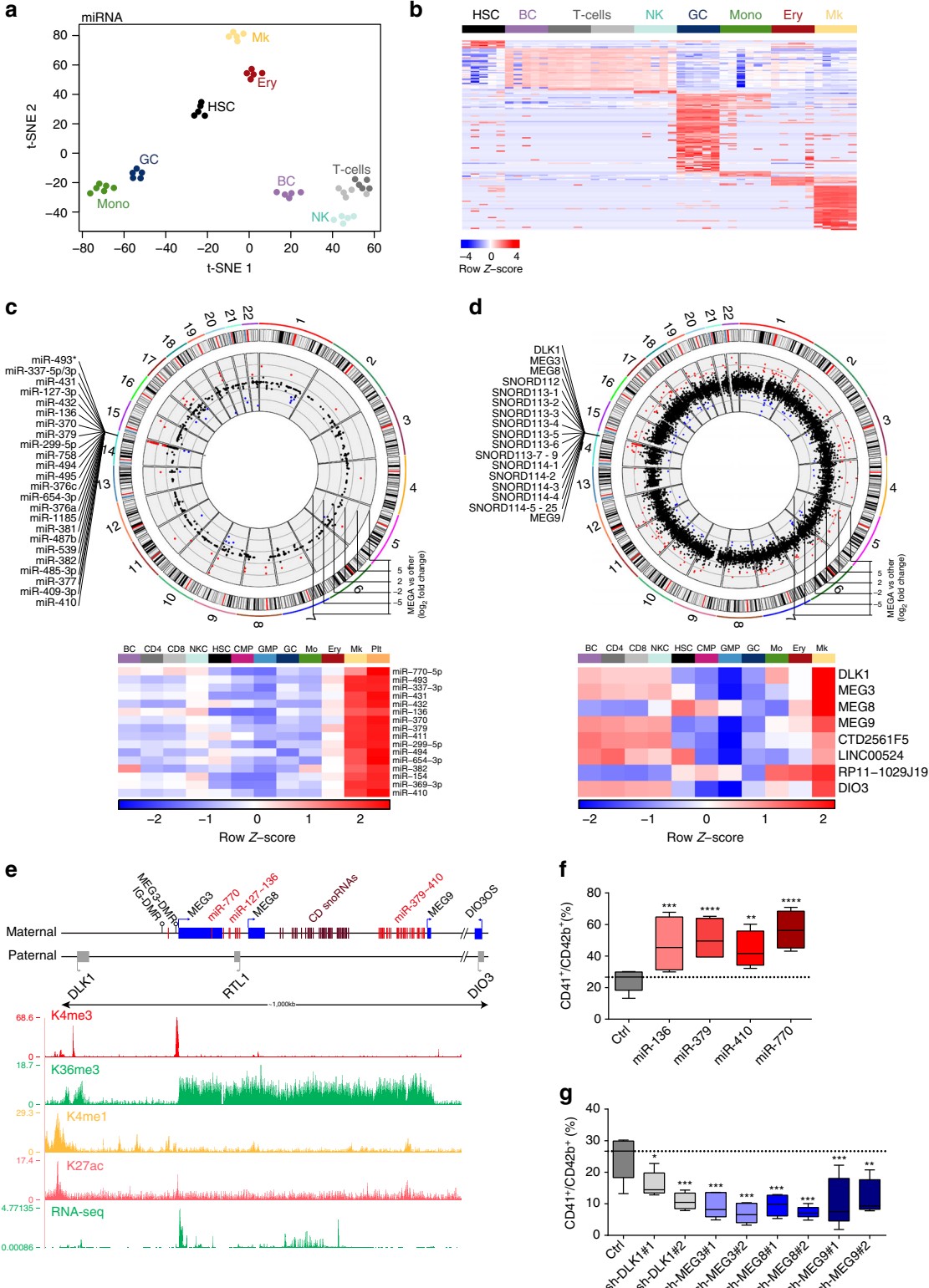

**Fig. 5** ncRNAs of the *DLK1-DIO3* locus control human megakaryopoiesis. **a** *t*-SNE of all samples using 240 variance-filtered miRNAs from the NCode Human miRNA Microarray V3 platform. **b** Heat map of 174 cell-type-specific fingerprint miRNAs. **c**, **d** Genome-wide view of log$_2$-FC for **c** miRNAs (NCode Human miRNA Microarray V3) and **d** ncRNAs (NCode Human Long Non-coding RNA microarray) in megakaryocytes compared to all other cell types. *Red dots*: log$_2$-FC ≥ 3. The heatmaps show qRT-PCR validation results of *z*-score transformed $2^{\Delta\Delta Ct}$ values. **e** Schematic of the imprinted *DLK1-DIO3* locus on human chromosome 14. Below are UCSC genome browser tracks (GRCh38/hg38) of ChIP-seq and RNA-Seq data in CD41$^+$ megakaryocytic cells[24]. **f** Lentiviral expression of miR-770, miR-136, miR-379, and miR-410 (n = 4) and **g** RNAi (shRNA)-mediated knockdown of *DLK1, MEG3, MEG8, and MEG9* (n = 5) in CD34$^+$ HSPCs in vitro. Both plots show the percentage of CD41$^+$/CD42b$^+$ cells. **f**, **g** Data are presented as mean ± s.e.m. *$P < 0.05$; **$P < 0.01$; ***$P < 0.001$; ****$P < 0.0001$; *ns* not significant; *P*-values were calculated using one-way ANOVA with Dunnett's post hoc test

whereas the SC-AML group II mostly consisted of *MLL*-rearranged samples (six out of eight samples). The remaining AML samples mapped to a space between the GMPs, monocytes and CMPs (referred to as group "MP-AML", Fig. 6b). Interestingly, pathway activity clustering and *t*-SNE revealed the effective absence of a myeloid expression program in SC-AML samples (Fig. 6c and Supplementary Fig. 8a). Thus, the SC-AML samples were not characterized by upregulation of HSC fingerprint ncRNAs compared to the remaining AML samples, but rather by downregulation of differentiation-associated ncRNAs (Fig. 6c and Supplementary Data 7). This suggests that transformed AML blasts express a conserved HSC program independent of the expression of a differentiation program, as has been previously shown for protein-coding genes[36, 37].

In order to uncover such a ncRNA HSC program in AML blasts, we proceeded with an integrated approach that could capture the shared features of AMLs and healthy HSCs. To this end, we trained a SOM containing 7094 ncRNAs selected by variance-based filtering. We identified two spots ("A" and "B"; Fig. 6d) containing 1215 and 300 ncRNAs, respectively, which were expressed in HSCs and AML blasts from diverse cytogenetic backgrounds (*MLL*-rearrangement and t[8;21]) and morphologies (with or without maturation; monoblastic/monocytic, or megakaryoblastic). The signatures could be further refined by calculating the overlap with limma, resulting in 576 spot A and 99 spot B high confidence ncRNAs, respectively (Supplementary Data 7). Pathway activity analysis revealed that samples with high expression of spot B ncRNAs were mainly from the core-binding factor rearranged subgroup, whereas samples with high expression of spot A ncRNAs came from all AML subgroups and showed upregulation of HSC fingerprint genes (Fig. 6e, f, Supplementary Fig. 8d, e and 9). Therefore, we reasoned that the ncRNAs in spot A are indeed HSC-related and represent a general ncRNA stem cell core signature.

**ncRNA stem cell signatures determine prognosis in AML.** To test whether our defined ncRNA stem cell signatures can be used to predict the survival of AML patients, we applied an unsupervised clustering approach and grouped 171 adult AML patients from an independent patient cohort[38] (Supplementary Table 1) based on their ncRNA expression (Fig. 7a, d). Indeed, patients who showed higher expression of the ncRNA stem cell signature (spot A signature) also showed significantly better overall (OS) and event-free survival (EFS; Fig. 7b, c). In contrast, downregulation of differentiation-associated ncRNAs—which characterized *t*-SNE SC-AML groups I and II (SC-AML signature)—predicted poor OS and EFS (Fig. 7d–f). Of note, the ncRNA expression signatures could add further prognostic value to a recently published gene stemness score that includes 17 coding genes (LSC17; Supplementary Fig. 10)[39]. Within the LSC17 high-risk patients (Supplementary Fig. 10a), the spot A signature and SC-AML ncRNA signature could identify those with a very high risk and an even worse OS (spot A: $P_{\text{log-rank}} = 0.011$; SC-AML: $P_{\text{log-rank}} < 0.0001$) and EFS (spot A: $P_{\text{log-rank}} = 0.006$; SC-AML: $P_{\text{log-rank}} < 0.0001$; Supplementary Fig. 10c, e). In contrast to the spot A lncRNA set, the SC-AML signature was also prognostic for patients with a low LSC17 signature (OS: $P_{\text{log-rank}} = 0.005$; EFS: $P_{\text{log-rank}} = 0.003$; Supplementary Fig. 10b, d, and f). Inversely, the average LSC17 score of patients grouped according to the SC-AML expression profile did not significantly differ (0.54 vs. 0.48; $P_{\text{Welsh's}} = 0.53$). The LSC17-high group did not contain a higher proportion of SC-AML high-risk patients than the LSC17 low group ($P_{\text{Fisher's exact}} = 0.79$), showing that the LSC17 score and our ncRNA signatures are indeed independent risk predictors. Thus, our expression resource enabled the identification of prognostically useful ncRNA signatures shared by normal HSCs and AML blasts of distinct cytogenetic and morphologic subgroups. These results show that the incorporation of ncRNAs will improve the prognostic value of published gene signatures based on coding genes.

**Discussion**

By high density reconstruction of the human coding and non-coding hematopoietic landscape, our study enabled us to identify highly relevant fingerprint ncRNAs that regulate lineage specification, HSPC maintenance and differentiation. Integration of a comprehensive set of pediatric AML samples allowed us to further define a core ncRNA stem cell signature in normal HSCs and AML blasts, which served as a prognostic marker in an independent cohort of AML patients. This signature will inform our understanding of self-renewal and the underlying transcriptional programs which are hijacked during malignant transformation, and may pave the way for novel therapeutic interventions targeting the non-coding transcriptome. The open-access resource provided by our study will be of value for advancing the current knowledge of ncRNA functions in normal hematopoiesis, and may help to uncover ncRNAs for therapeutic targets in myeloid leukemia and for regenerative medicine.

For the granulocyte-specific *LINC00173*, we demonstrated that our expression resource coupled with the established bioinformatic pipeline can define functionally relevant fingerprint lncRNAs and infer their functions. We validated that *LINC00173* controls the proliferation of myeloid progenitor cells and differentiation into granulocytes. We propose that this occurs through transcriptional silencing of the expression programs of alternative blood lineages via interaction between *LINC00173* and EZH2, a core component of the repressive PRC2 complex. Future studies will need to investigate whether modulating *LINC00173* expression can instruct or block granulocytic differentiation in patients with perturbed granulopoiesis, such as severe congenital neutropenia (Kostmann's disease). As we have made our fingerprint ncRNAs plus their co-expressed coding gene sets publically available in an online resource, we expect our data set to prompt the discovery and characterization of many more functional ncRNAs with roles in health and disease.

Our work additionally underscores the importance of studying ncRNAs in human samples in order to understand and perturb the human blood system. While Qian et al. showed that ncRNAs from the imprinted *Dlk1-Gtl2* cluster are specifically expressed in mouse LT-HSCs and are essential for their maintenance[7], we did not observe this specificity in the human system. Instead, miRNAs, snoRNAs and several lncRNAs from the human *DLK1-DIO3* locus were highly expressed in megakaryocytes and controlled the differentiation of these cells. One possible explanation for the prominence of this cluster in both HSCs and megakaryocytes is the close relationship between the two cell types[40]—underlined by the recent proposal that human megakaryocytes differentiate through a direct route from HSCs, bypassing a multipotent progenitor cell state (i.e., megakaryocytic/erythroid progenitor cells)[41]. Whether the miRNAs of the human *DLK3-DIO3* locus also maintain megakaryopoiesis through repression of PI3K-mTOR signaling—as shown for LT-HSCs[7]—remains to be elucidated, as does the mechanism through which the lncRNAs feed into this circuit.

Despite the enormous diversity of reported genetic alterations in AML[38] affecting many different cellular pathways and programs, the ultimate result of their interplay in AML blasts is uniform. Malignant stem or progenitor cells possess enhanced self-renewal capacity while their differentiation is abnormal[36].

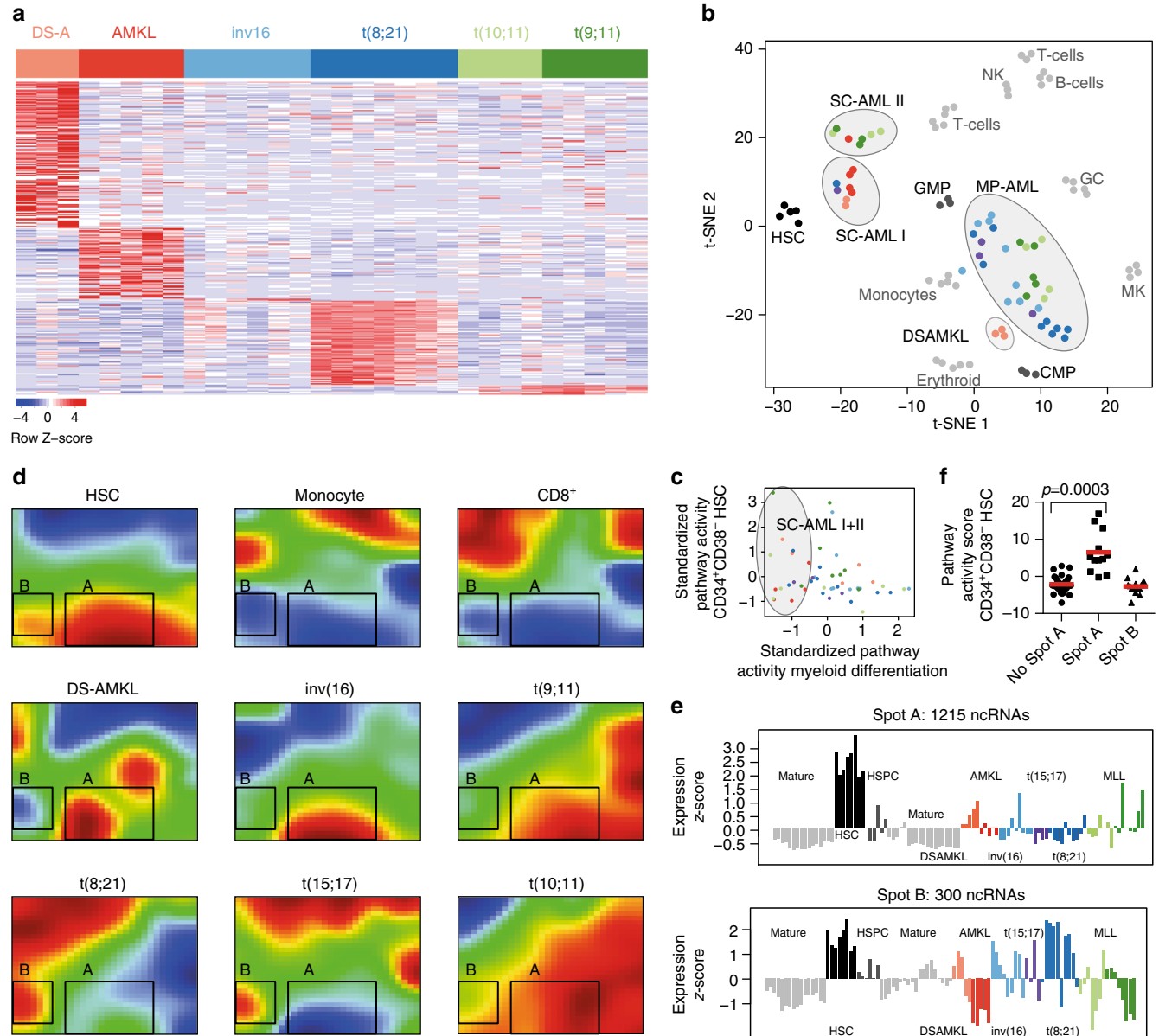

**Fig. 6** Integrated analysis reveals ncRNA stem cell signature in AML and HSCs. **a** Heatmap of pediatric AML subtype-specific fingerprint ncRNAs. **b** Two-dimensional *t*-SNE plot using 1877 variance-filtered ncRNAs. The clusters were called by densVM and correspond to 11 healthy populations and 4 AML clusters. **c** Standardized pathway activity scores[34] of the myeloid (*x*-axis) vs. HSC-specific expression program (*y*-axis) as defined by our resource. Samples belonging to the SC-AML I and II clusters are indicated by gray ellipses. **d** SOM-expression portraits obtained using 7094 variance-filtered ncRNAs as input. Shown are three healthy cell populations (*top*) including HSCs (*top left*)—which are characterized by high expression of Spot A and B metagenes (corresponding to 1215 and 300 ncRNAs, respectively)—and AML samples displaying high expression of Spot A or Spot B metagenes. All samples are shown in Supplementary Fig. 9. **e** Expression profiles of Spot A metagenes (above) and Spot B metagenes (below) across all samples. **f** Pathway activity scores[34] for the HSC-specific expression programs in "Spot A" and "Spot B" AMLs. *P*-value was calculated using the two-tailed Welsh's *t*-test

Therefore, we reasoned that normal HSCs and AML blasts of different cytogenetically and morphologically defined subgroups share a common characteristic, namely a core self-renewal or stemness program[37]. We applied artificial neural networks to identify such a core ncRNA stem cell signature. Since lncRNAs have the capacity to precisely control the cellular epigenetic and transcriptional landscape[30], this core ncRNA signature may be essential for establishing and maintaining a stem cell-like state. Indeed, we showed that high expression of the ncRNA signature or the absence of a ncRNA differentiation program serve as prognostic factors in an independent cohort of 171 adult AML patients, suggesting the functional importance of these ncRNAs.

As these signatures added prognostic value to a previously published 17 coding gene stemness score in AML[39], we believe that ncRNAs expression data will help to identify patients who are at high risk of death or relapse and may profit from more intensive or alternative treatment approaches.

As of yet, therapeutic targeting of causative oncogenic proteins has remained widely unsuccessful. Hence, unraveling the self-renewal programs shaped by lncRNAs will not only enhance regenerative medicine. Defining similarities and differences between stem cell programs in normal HSCs and leukemic cells will also delineate a yet unrecognized therapeutic window, enabling us to develop novel cancer-specific treatments. With

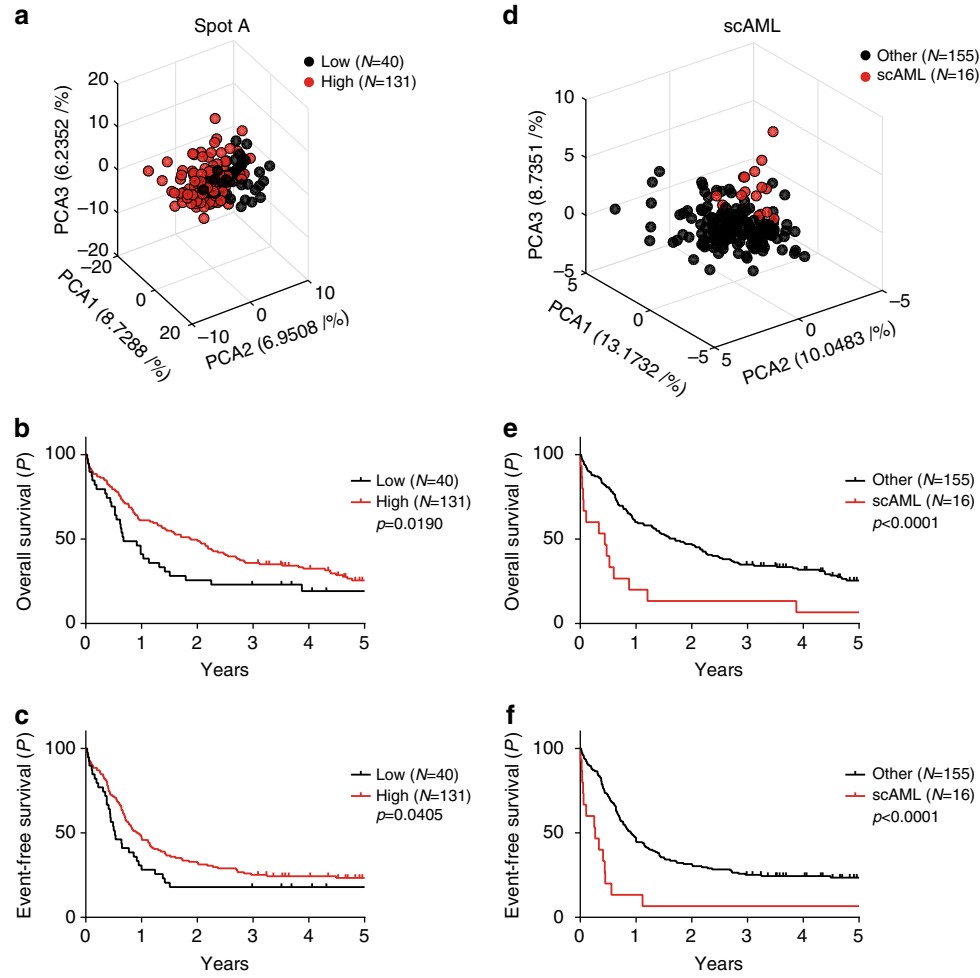

**Fig. 7** ncRNA stem cell signatures predict survival in adult AML. **a** PCA of 171 adult AML samples[38] using the Spot A signature (Supplementary Data 7). **b** Kaplan–Meier curves of 5-year overall survival and **c** event-free survival of AML patients, grouped via unsupervised k-means clustering according to their expression of Spot A ncRNAs. **d** PCA of adult AML samples using the differentiation-associated SC-AML signature (Supplementary Data 7). **e** Kaplan–Meier curves of 5-year overall survival and **f** event-free survival of the stem cell AMLs (SC-AML) in comparison to the other AMLs, as separated by unsupervised k-means clustering. *P*-values were calculated using the log-rank test

recent advances in RNA-based therapies, targeting aberrant transcriptional programs in AML is a strategy that is now within reach[42]. Our publically available resource and bioinformatic pipeline will certainly advance our understanding of the transcriptional organization underlying stem cell homeostasis and lineage specification, and will bring us another step closer towards achieving this overarching aim.

## Methods

**Primary cells and patient samples.** Cord blood (CB), peripheral blood (PB), and G-CSF mobilized mononuclear cells (PBMCs) were obtained from anonymous healthy donors. HSPCs were positively selected by labeling CD34-expressing cells with magnetic cell-sorting beads (Miltenyi Biotech). Fetal liver CD34$^+$ HSPCs were purchased from Novogenic Laboratories (LLC USA). Culture conditions for maintenance, granulocytic, megakaryocytic, erythroid, and megakaryocytic/erythroid in vitro differentiation of CD34$^+$ HSPCs have all been previously described[43, 44]. Megakaryocytes and erythroblasts were obtained from CB CD34$^+$ HSPCs through 7 days of in vitro differentiation followed by sorting for CD41$^+$/CD42b$^+$ (megakaryocytes) or CD36$^+$/CD235a$^+$ (erythroid cells). Granulocytes (FS$^{high}$, SS$^{high}$, CD15$^+$, CD66$^+$), monocytes (FS$^{high}$, SS$^{low/mid}$, CD14$^+$), NK cells (FS$^{low}$, SS$^{low}$,CD56$^+$,CD3$^-$), CD4-T-cells (FS$^{low}$, SS$^{low}$, CD3$^+$, CD4$^+$, CD8$^-$), CD8-T-cells (FS$^{low}$, SS$^{low}$, CD3$^+$, CD4$^-$, CD8$^+$), and B-cells (FS$^{low}$, SS$^{low}$, CD19$^+$, CD3$^-$, CD56$^-$) were sorted based on surface markers from the PB of five healthy donors. HSCs (Lin$^-$/CD34$^+$/CD38$^-$), CMPs (Lin$^-$/CD34$^+$/CD38$^+$/CD123$^+$/CD45RA$^{mid}$), and GMPs (Lin$^-$/CD34$^+$/CD38$^+$/CD123$^+$/CD45RA$^{high}$) were sorted from CD34-enriched HSPCs from fetal liver samples, CB, or PBMCs. Cell lines (THP-1 (DSMZ: ACC-16), NB4 (ACC-207), 293T (ACC-635), HT1080 (ACC-315), 32D

(ACC-411) were purchased from the German National Resource Center for Biological Material (DSMZ), maintained under recommended conditions and monthly tested negatively for mycoplasma. The Berlin-Frankfurt-Münster AML Study Group (AML-BFM-SG, Hannover, Germany) provided anonymous AML patient samples. Bone marrow or PB samples were sorted for blasts (AMKL/DS-AMKL: CD3$^-$, CD19$^-$, CD41$^+$, CD117$^+$, and/or CD34$^+$; other AML subtypes: CD3$^-$, CD19$^-$, CD117$^+$, and/or CD34$^+$). Informed consent was obtained from all human participants. All investigations were approved by the local Ethics Committee of Hannover Medical School and were performed in accordance with the declaration of Helsinki and local laws and regulations.

**Microarray data collection.** Total RNA was isolated with Quick RNA Microprep for HSCs, CMPs, GMPs, MEPs, and AML samples with <1×10$^5$ cells and with Quick RNA Miniprep for all other samples (both Zymo Research). RNA quality was assessed using the Agilent 2100 Bioanalyzer. For microarray analysis, the Agilent Array platform was employed. Briefly, rRNA was removed from 1 μg total RNA (mRNA-ONLY Eukaryotic mRNA Isolation Kit, Epicentre). Each sample was then amplified and transcribed into fluorescent cRNA using a random-priming method, thus capturing the entire length transcripts without 3′ bias. The labeled cRNA was hybridized onto three platforms: the Arraystar Human lncRNA Microarray V2.0 (Agilent-033010), and the NCode Human Long Non-coding RNA microarray (Agilent-021441) and NCode Human miRNA Microarray V3 (Agilent-021827). After washing the slides, the arrays were scanned by the Agilent Scanner G2505B. Agilent Feature Extraction software (version 10.7.3.1) was used to analyze acquired array images. These steps were either performed by Arraystar Inc. or the core facility of the Helmholtz Center for Infection Research in Braunschweig, Germany. For *LINC00173* knockdown in human CD34$^+$ HSPCs, SurePrint G3 Human Gene Expression v3 Microarrays (Agilent-072363) were used. These were processed in the core facility of the Helmholtz Center for Infection Research in

Braunschweig, Germany, using the Agilent Technologies Scanner G2505C. Back-ground corrected features were extracted using Agilent Feature Extraction software.

The data were analyzed using R and Bioconductor[45]. Arraystar Human lncRNA Microarray V2.0 NCode Human Long Non-coding RNA microarray, and NCode Human miRNA Microarray V3 data sets were processed and analyzed separately, with the exception described in Supplementary Fig. 1a–c for PCA on a joint data set of 15,219 GENCODE-annotated transcripts shared by the Arraystar Human lncRNA Microarray V2.0 and NCode Human Long Non-coding RNA microarray platforms. For assessment of the global concordance between the two ncRNA platforms, the correlation coefficients between all possible pairwise sample combinations on each platform were calculated and plotted against each other, yielding a "correlation of correlations" coefficient[46].

**Probe annotation**. The GENCODE v23 (release 09/2015)[47], LNCipedia 3.1 (release 02/2015)[48], and NONCODE v4 (release 01/2014)[49] transcript databases were downloaded as FASTA files. All 60mer probes on each array (Arraystar Human lncRNA Microarray V2.0: 60,699, NCode Human Long Non-coding RNA microarray: 39,246) were then aligned against primary transcript sequences with one mismatch allowed. In the case of multiple hits, final gene symbols, and names were assigned in the following order of priority: 1. GENCODE, 2. LNCipedia, 3. NONCODE, 4. as designated by the manufacturer including more than 4000 lincRNAs described by Cabili et al[4]. For the NCode Human miRNA Microarray V3 the manufacturers annotation was used.

**Preprocessing**. Array quality was checked by inspection of probe intensity distributions and by PCA of log2-transformed unprocessed data. At this point three outlier arrays (1 B-cell, 1 NK-cell, and 1 CD4-T-cell array from the Arraystar set) were excluded from further analysis due to low signal strength. Data were log2-transformed and quantile-normalized using the "limma" package[15]. For the integrated analysis of healthy blood and AML samples, a batch correction between AML and healthy cells was performed on quantile-normalized log2-values, using the parametric ComBat algorithm as implemented in the R package sva[50].

**Selection of probes for unsupervised analyses**. The performance of unsupervised (class discovery) algorithms depends on the number of features used as input. To reduce the noise inherent to any high-dimensional data set, selection of potentially interesting features displaying at least some variation needs to be applied. This is normally performed using the interquartile range or coefficient of variation (standard deviation (s.d.)/mean). We applied both methods to our data set using the R package genefilter. However, upon close inspection of the selected features we noted that many informative genes were not selected, either due to exclusive expression in a subgroup of samples constituting <12.5% of the total sample size or due to relatively high baseline expression in all other samples. To circumvent this problem and to select features that might be expressed highly specifically in our smallest subgroups (NK cells; CMPs, GMPs: three samples each) we used an adaption of the ROSE algorithm (Recognition of Outliers by Sampling Ends)[51]. For ROSE, the intensity values of each probe were plotted in ascending order over all samples. The plots were divided into thirds and a linear regression curve was fitted to the middle third of the set of intensities. The third and ante-penultimate samples (to account for group size ≥3) were used as fixed cutoff points were the observed intensity was compared to the trend line. When the observed log2-FC at these points was higher than a defined threshold the probe was included for further analysis. This procedure excluded probes showing only high variances and selected a higher number of informative, lineage-specific "fingerprint" genes. The R-script for our implementation of ROSE is available upon request. The thresholds used for ROSE were as follows: for Arraystar Human lncRNA Micro-array V2.0, Fig. 2a–c t-SNE: log2-FC > 2 and log2-FC < 2.5 resulting in 7077 probes; Fig. 2d SOM: log2-FC > 1 and log2-FC < 2 resulting in 17,655 probes; and for NCode Human Long Non-coding RNA microarrays, Supplementary Fig. 2a–c t-SNE: log2-FC > 1.5 and log2-FC < 1.7 resulting in 5279 probes; Supplementary Fig. 2d SOM: log2-FC > 0.6 and log2-FC < 1.5 resulting in 14,256 probes.

**Unsupervised analyses**. For 3D and 2D representation of sample maps we used t-SNE[13]. t-SNE is a non-linear dimensionality reduction technique which maps the original sample to sample distances in the high-dimensional feature space. These distances are subsequently mapped to a lower dimensional space by stochastic minimization of the Kullback–Leibler divergence between the distribution of pairwise similarities in the high-dimensional space and the t-distribution of similarities in the lower dimensional space. We used the Barnes–Hut implementation of t-SNE from the Rtsne-package on the indicated numbers of ROSE-selected probes with no prior PCA, and visualized the results as 2D or 3D plots. Density-based clustering for automated subset identification was performed on the BH-SNE output using the function densVM from the R-package cytofkit[35]. PCA was carried out with the prcomp function from the R-package stats. Heatmaps were generated using the function heatmap.2 from the R-package gplots. Unless otherwise noted all heatmaps were row-scaled with the color key indicated below the heatmap.

**Self-organizing maps**. Self-organizing maps allow dimensionality reduction, similarity analyses and easy extraction of group-specific co-regulated genes[14]. During the training process of a SOM every gene is assigned to a representative metagene that matches its expression profile across the data set. Each metagene is represented by a tile in a mosaic grid (in our case 30 × 30). The SOM is trained in such a way that metagenes with similar expression profiles localize to the same region of the SOM, thereby forming spots of co-expressed genes. For the Arraystar Human lncRNA Microarray V2.0 and NCode Human Long Non-coding RNA microarray SOMs we used 17,655 and 14,256 ROSE-selected probes (mRNAs and ncRNAs), respectively, as inputs to the oposSOM package[14]. The algorithm was run with the groupmap parameter set to 0.85; otherwise the default parameters recommended in the package vignette were used. The phylogenetic trees were reconstructed by the oposSOM package using neighbor-joining on spot metagenes identified by the package.

**Fingerprint genes**. We used the limma package[15] with Benjamini–Hochberg multiple-testing correction to detect differentially expressed probes. Every lineage-specific "fingerprint" gene was required to show a significant moderated t-test statistic (adjusted P < 0.05 and log2-FC > 1) in the respective lineage vs. every other lineage in the data set, and additionally to be significantly enriched in the corresponding group overexpression spots in the SOM.

**"Guilt-by-association" and gene set enrichment analyses**. The guilt-by-association approach for ncRNAs was carried out as previously described[16]. Briefly, a Pearson product moment correlation matrix was constructed between over 4000 signature ncRNAs (from the fingerprints and SOM overexpression spots) and 18,295 unique protein-coding genes. This resulted in 18,295 correlation coefficients for each investigated ncRNA, constituting a ranked gene list. These ranked lists were then used as inputs for PAGE[17], which is significantly faster than the classical GSEA algorithm[18] since it uses the normal distribution to infer statistical significance[17]. We used the PAGE implementation from the Bioconductor package piano with the standard parameters recommended by the authors. The FDR was estimated using the standard Benjamini and Hochberg approach on nominal P-values. The gene sets tested for enrichment were from MSigDB.v5.1 (C2, C3, C5, C6, hallmark gene sets) and 140 custom gene sets related to hematopoiesis. The latter are given in Supplementary Data 3. Gene sets smaller than 15 genes or larger than 300 were filtered out, resulting in 5591 gene sets that were tested for each of the over 4000 ranked gene lists. The enrichment results for HOTAIRM1 and LINC00173 were visualized using the output from the BROAD GSEA software[18] as the input for the Enrichment Map plugin[52] of Cytoscape 3.3.0. Gene sets with a nominal P < 0.001 and a FDR < 0.05 (based on a simulated null distribution of 1000 random gene set permutations and multiple-testing adjustment to control the FDR as computed by the BROAD GSEA software) were selected for visualization. For validation of the guilt-by-association pipeline two independent AML data sets (GSE15434 and GSE14468)[21, 22]; both on the Affymetrix Humane Genome U133 2.0 Plus platform) were downloaded from GEO including the associated metadata. We performed quality control and discarded outlier samples based on the raw intensity distribution of the probesets and normalized unscaled standard error (NUSE) plots, which yielded n = 190 CN-AML samples in GSE15434 and n = 457 AML samples in GSE14468, including n = 187 CN-AML. The guilt-by-association approach for HOTAIRM1 (228642_at) was repeated as described above.

For cross validation of our top lineage-specific genes and leading-edge genes with the human DMAP data set[12], the DMAP data set was downloaded from the GEO (GSE24759), preprocessed using RMA[53], and the top lineage-specific genes were extracted for each lineage. A custom gene set database (Supplementary Data 3)—comprised of our fingerprint coding genes and the top lineage-specific genes from the DMAP—was constructed and tested on the Arraystar, NCode, and DMAP data sets using the Broad GSEA tool. Gene sets smaller than 15 genes or larger than 300 were filtered out, the data sets were collapsed to contain only unique identifiers, and the permutation type was set to Gene_set (1000 permutations). The other parameters were all set to default values.

For the LINC00173-knockdown experiments in primary human CD34+ HSPCs cultured under myeloid differentiation conditions, three independent biological replicates were preprocessed separately by quantile normalization with the limma package[15]. The Agilent SurePrint G3 Human GE v3 expression matrix (58,341 features) was filtered to contain only coding genes. In cases with multiple probes per gene the most variable probe was selected, resulting in 18,295 unique coding genes. log2-FC between knockdown and control samples within each replicate were computed for all unique 18,364 coding genes using the limma package. These 18,364 log2-FC values were subsequently analyzed with the Broad GSEA software[18] using GSEA-preranked with the permutation type set to Gene_set (1000 permutations). GSEA results were averaged over the three replicates.

Assessment of pathway activity in single AML samples was performed using a single-sample gene set enrichment method which aggregates z-score transformed gene expression values of a given gene set into a single pathway activity score[34]. The algorithm is implemented in the R-package GSVA[54]. Briefly, the Arraystar expression matrix (60,699 features) was filtered to contain only coding genes. In cases with multiple probes per gene the most variable probe was selected, resulting in 18,295 unique genes. The algorithm transforms the gene-by-sample matrix into a gene set-by-sample matrix. For the gene set database we used 80 custom gene sets

related to hematopoiesis and leukemia. The 30 gene sets with highest variance were selected for the heatmap shown in Supplementary Fig. 8a.

**Circos plots**. The megakaryocyte vs. all other contrast was visualized by plotting limma-generated log$_2$-FC values over genomic locations using the R-package RCircos[55]. log$_2$-FC > 3 were indicated by red and log$_2$-FC < 3 were indicated by blue in the Circos plots.

**RNA-seq data collection and analysis**. RNA from sorted healthy bone marrow subpopulations was obtained as previously described[23]. The sample quality was verified with the DNA 1000 Assay on an Agilent Bioanalyzer. Subsequently, libraries were generated using the TruSeq Stranded Total RNA LT (with Ribo-Zero Gold) kit from Illumina and sequenced on an Illumina HiSeq at the Stanford Sequencing Service Center, CA, USA. RNA-seq reads (on average $85 \times 10^6$ read counts per sample) were aligned to the human reference genome GRCh38 release 23 using STAR version 2.4[56]. Alignment quality statistics and count matrices were computed using HTSeq[57] by mapping the reads to the exons of GENCODE.V23[47]. All uniquely aligned reads that were unambiguously assigned to annotated exons were submitted to further expression analysis with DESeq2[58]. SOMs were trained by using rlog-transformed (Regularized logarithm transformation) values of normalized counts as input for the oposSOM package[14]. ncRNAs which mapped to the SOM-modules were then probed for differential expression with DESeq2 using pairwise comparison between all lineages. To compare the coverage of RNA-seq, NCode Human Long Non-coding RNA microarrays, and the Arraystar Human lncRNA Microarray V2.0 microarrays, we analyzed expression sets of similar populations of cells (HSPC/HSC plus granulocytes). For estimation of the real coverage we modeled the background distribution on the arrays by fitting a Gaussian kernel to the lowest 15% of the probes forming the peak at the lower end of the probeset distribution. Genes were marked as expressed if the signal was above the mean background plus three s.d. on more than four arrays. For the RNA-seq data mapped to GENCODE.V23 we filtered out all features with a summarized expression of <15 reads in total over the nine samples.

**Analysis of TCGA AML RNA-seq data**. RNA-seq data were obtained from the AML project of The Cancer Genome Atlas Research Network[38]. The raw sequencing files were aligned to the human genome (hg38) using TopHat2 (v2.011)[59]. Experiment-specific parameters (e.g., read lengths or inner-mat distances) were estimated from each data set independently, while all other parameters were those suggested by the developers. Coding and non-coding genes were annotated using GENCODE[47], LNCipedia[48], NONCODE[49], and the lincRNA catalog[4], and expression levels were quantified as sequencing reads across exons using HTSeq (v0.6.1)[57]. Visualization of these data identified a sample outlier, TCGA-AB-2811, which we excluded from further analysis. The expression of gene signature $i$ was thus defined as $E_i = [e_{n,m}]$, $n = 1,…,N_i$ and $m = 1,…,M$ where $N_i$ is the number of genes in signature $i$ and $M = 171$ is the number of patient samples analyzed in this study. The matrix $E_i$ was normalized using the trimmed mean approach[60] and unsupervised k-means clustering was applied to optimally separate all 171 patient samples into two groups[37]. The Kaplan–Meier and Cox-regression models were then used to determine whether OS and EFS were significantly different between the two groups.

**TCGA AML microarray data and LSC17 score calculation**. Microarray data of 183 AML patients within the TCGA LAML data set[38] were obtained from the TCGA data portal. Raw Affymetrix CEL files (generated on the HG-U133 Plus 2.0 array) were processed with the R package gcrma. The LSC17 score was calculated for each patient as described[39] as a linear combination of Affymetrix probesets belonging to the LSC17 signature with the provided coefficients for each probeset. The median of all LSC17 scores was determined and patients were assigned a LSC17-high status if they had a LSC17 score above the median or LSC17 low if their LSC17 score was below the median. For analysis of overlaps between the ncRNA signatures and LSC17 scores, only patients who had Microarray and mapped RNA-seq data available were included (162 patients). The survival between the different groups was compared using Kaplan–Meier Plots and standard log-rank tests.

**Transduction and hematopoietic assays**. CD34$^+$ HSPCs were lentivirally transduced on RetroNectin-coated (Takara) plates as previously described[6], and sorted according to the construct's fluorescent marker (Cerulean or eGFP). Methylcellulose-based (Human Methylcellulose Complete Medium HSC003, R&D Systems) and collagen-based (Megacult, Stem Cell Technologies) colony-forming assays were carried out according to the manufacturers' instructions and analyzed on day 14. Five thousand sorted cells were used as input. CD41$^+$ cells on Megacult assays (Stem Cell Technologies) were enumerated by scoring stained colonies against the Evan's Blue counterstain. Slides were imaged on a BZ9000 (Keyence) automated microscope, and merging, background fluorescence reduction, gamma-level, brightness and contrast enhancement, and counting by Hybrid-Cell-Counter were again performed using BZ-II Analyzer v.2.2 (Keyence).

**Cytochemistry and cell assays**. Standard protocols were used to perform May-Grünwald Giemsa staining of cytospins. Myelocytic metabolic activity was analyzed using immunohistochemical leukocyte POX staining. Briefly, cytospins of cells collected fixated for 30 s using methanol with 1% formalin. After rinsing in water, the slides were incubated for 12 min in filtered staining solution (160 mg $C_{14}H_{14}N_2$ in 4 ml acetone, 4 ml DMSO, and 0.03% $H_2O_2$; all Sigma), followed by a second water rinse. The slides were then dried at room temperature and stained with Haemalaun for 20 min. After a final water rinse, they were fixated using Kaiser's glycerin gelatin (Merck). Cell growth was quantified by trypan blue exclusion dye, or by flow cytometry-based cell counting using a CytoFLEX flow cytometer (Beckman Coulter). Growth competition assays were performed by lentiviral supernatant infections reaching transduction efficiencies between 20–40%, thus yielding a mixed population of construct-positive fluorescent cells and untransduced competitor cells. All subsequent measurements were normalized to day 0. Phagocytosis capacity was assessed on day 12 by culturing cells for 24 h in RPMI (with 10% FCS and 1% Penicillin/Streptomycin). Amine-modified polystyrene latex beads of 2 μm mean particle size (Sigma) were added to the cells at a ratio of 1:2000. After 3 h of incubation at 37 °C (a negative control was incubated at 4 °C), the cells were washed three times and bead uptake was measured by flow cytometry.

**Lentiviral constructs**. shRNAs against human lncRNAs were obtained by applying the SENSOR design rules[61] and subcloning the 97mer oligos into a pLKO5d.SFFV. eGFP.miR30n backbone construct (Addgene #90333). A non-silencing shRNA against Renilla luciferase was used as a control (sh-ctrl). For shRNA reporter assays, gBlocks (Integrated DNA Technologies, Inc., IDT) with shRNA binding sites were inserted into pTtNPT or pRSF91.mTagBFP2.Sensor.WPRE as described[62] to generate stable 32D reporter cell lines, which were then transduced with shRNA constructs to perform the reporter assay[62]. For CRISPRi experiments the lentiviral construct pLKO5d.SFFV.dCas9-KRAB.P2A.BSD (Addgene #90332) was used to generate stable NB4 cell lines. sgRNA oligos were designed using CCTop[63] and selected based on CRISPRi design rules[29], then subcloned into a pLKO5 derivative with a fluorescent color (RFP657) and the human U6 promoter for driving sgRNA expression. A non-silencing sgRNA against Renilla luciferase was used as a control (Luc). Human miRNA genes were cloned into LeGO-vector derivatives and lentiviral supernatant was generated and collected using standard protocols as previously described[64]. All shRNA and sgRNA sequences are shown in Supplementary Data 8.

**Flow cytometry and cell sorting**. Transduced HSPCs were sorted based on expression of GFP or Cerulean. In vitro differentiation was analyzed on day 14 of culture. Flow cytometry analyses were performed on a Navios 10/3 or a CytoFLEX B5-R3-V5 (both Beckman Coulter). Kaluza 1.3/1.5 (Beckman Coulter) or FlowJo V10 were used for data analysis. Staining and measurement were performed according to standard protocols as previously described for human cells[44], using the antibodies FITC-CD8 (B9.11), FITC-CD19 (89B), FITC-CD38 (T16), FITC-CD41 (P2), FITC-CD66b (80H3), PE-CD42b (SZ2), PE-CD56 (IM2073U), PE-CD123 (9F5), PE-CD117 (95C3), PC5.5-CD14 (RMO52), PC7-CD3 (UCHT1), PC7-CD34 (581), PC7-CD41 (P2), PC7-CD235a (11E4B-7-6), APC-CD4 (13B8.2), APC-CD13 (Immu103.44), APC-CD34 (581), APC-CD45RA (2H4LDH11LDB9), AlexaFluor750-CD19 (89B), AlexaFluor750-CD235a (11E4B-7-6), KromOrange-CD3 (UCHT1) (all Beckman Coulter), PE-CD36 (CB38), PC7-CD66b (G10F5), and APC-CD42b (HIP1) (Becton Dickinson).

**Quantitative real-time PCR**. Total RNA was isolated with Quick RNA Microprep for HSCs, CMPs, GMPs, and AML samples with $< 1 \times 10^5$ cells and with Quick RNA Miniprep for all other samples (both Zymo Research). cDNA synthesis was done using the High Capacity cDNA Reverse Transcription Kit (Applied Biosystems) with 500 ng or 1 μg input RNA depending on the sample group. QRT-PCRs were performed using SYBR Select Mastermix (Thermofisher). QRT-PCR primer sequences are available upon request. MiRNA-detection was performed with TaqMan miRNA assays (ABI). All data were measured in a StepOnePlus Cycler (ABI). Microarray validation ΔCt's were calculated as $2^{-(Ct_{GOI}-Ct_{HK})}$ (GOI, gene of interest; HK, housekeeping gene). Overexpression and knockdown fold changes were quantified using the geNORM ΔΔCt equations. RNA fractionation into cytoplasmic and nuclear lysates was done using the Cytoplasmic & Nuclear RNA Purification Kit (Norgen biotek corp.) according to the manufacturer's instructions. Expression profiling in the different compartments was performed by qPCR as described above. Ratios were calculated as $2^{-(Ct_{cytoplasmic}-Ct_{nuclear})}$.

**RNA fluorescence in situ hybridization**. Probes were designed using Stellaris probe designer 4.0 (Bioisearch technologies). For *LINC00173* a masking level of 5 was applied, retrieving 26 probes used at a concentration of 3 μM. Controls were predesigned probesets and were used at 250 nM (GAPDH) and 500 nM (MALAT1). For RNA FISH, THP-1 cells, or CD34$^+$ PB HSPCs differentiated for 11 days in granulocytic differentiation medium were used. These were washed and suspended at $10 \times 10^6$ ml$^{-1}$ in phosphate-buffered saline (PBS), of which $1 \times 10^6$ cells were plated on a Poly-L-Lysine coated cover glass. After PBS evaporation, the cover glass was placed into a 6 well plate and fixed with 3.7% formaldehyde in

1×PBS for 10 min. Following a PBS wash the cells were permeabilized using 70% ethanol for 1 h at +2 to +8 °C, then washed using wash buffer (10% formamide in 2×SSC; Biosearch technologies). For the hybridization step the probes were diluted in hybridization buffer (100 mg ml⁻¹ dextran sulfate and 10% formamide in 2×SSC) and dispensed onto the cells. The solution was covered with Parafilm and the cover glass was placed into a humidified chamber (150 mm tissue culture plate with a flat water-saturated paper towel and a single layer of Parafilm placed on top of the paper towel, all covered with a tissue culture lid and sealed with Parafilm) for 8 h of incubation at 37 °C. The cells were then washed with wash buffer and incubated in the dark for 30 min, followed by nuclear staining with 5 ng ml⁻¹ DAPI in wash buffer and another 30 min in the dark, and finally a wash with 2×SSC. For mounting, Glox buffer anti fade (0.4% glucose in 10 mM Tris, 2×SSC with 3.7 mg ml⁻¹ glucose oxidase, and catalase suspension (SIGMA)) was dispensed onto the cells and covered with a microscope slide, and the edges were sealed with nail polish. Imaging proceeded using a BZ9000 (Keyence) automated microscope. Background fluorescence reduction, contrast enhancement, and merging were performed with Biorevo Software (Keyence).

**RNA immunoprecipitation**. RIP was performed as previously described[65]. Briefly, $1 \times 10^7$ NB4 or THP-1 cells were lysed and frozen at −80° overnight. Dynabeads protein G (Invitrogen) were then washed twice using a Dynal magnet (Invitrogen) with 0.5 ml citrate-phosphate buffer (pH 5), and resuspended in citrate-phosphate buffer with the following antibodies: Anti-EZH2 polyclonal rabbit antibody (07-689; Millipore), Anti-Ezh2 (D2C9) XP rabbit antibody (Cell Signaling), and rabbit (DA1E) mAb IgG XP Isotype Control #3900 (Cell Signaling). Reactions were left at room temperature while rotating for 30 min. The beads were washed and mixed with cleared (14,000×g for 10 min at 4 °C) lysate (100 µl) overnight at 4 °C. Afterwards, the beads were washed six times with 500 µl of cold NT-2 Buffer. Then beads were resuspended in 300 ml Proteinase K Buffer (Invitrogen) and shaken (300 rpm) at 55 °C for 30 min. RNA was isolated using phenol:chloroform:isoamyl and precipitated with 50 µl 5 M ammonium acetate, 15 µl 7.5 M lithium chloride, 5 µl of 5 mg ml⁻¹ glycogen and 850 µl absolute ethanol per reaction. After centrifugation and washing with 80% ethanol, RNA was air dried, and resuspended in 20 µl RNAse free water. For cDNA synthesis 500 ng of RNA was used.

**Chromatin Immunoprecipitation**. ChIP assays were performed as previously described[66] with $1 \times 10^6$ cells per condition using antibody against H3K27me3 (Millipore, 07-449). Briefly, cells were collected and washed with PBS, followed by incubation in 1% (w/v) formaldehyde for 10 min at room temperature. To terminate cross-linking reaction, cells were incubated with 0.125 M glycine for 5 min while rotating. Cells were washed with PBS and lysed on ice in cell lysis buffer (10 mM Tris [pH 8.0], 10 mM NaCl, and 0.2% NP-40) for 10 min to recover nuclei. After centrifugation at 1500×g for 5 min, nuclei were lysed in nucleus lysis buffer (50 mM Tris, 10 mM EDTA, and 1% SDS [pH 8.0]) on ice for 10 min. The lysate was diluted in IP dilution buffer (20 mM Tris [pH 8.0], 2 mM EDTA, 150 mM NaCl, 1% Triton-X100, and 0.01% SDS) and sonicated (Settings: High, 30 s pulses, 12 cycles) using a BioRuptor Pico sonicator (Diagenode, Liège, Belgium) to yield an average fragmentation size of ~200 bp. The chromatin was precleared with 5 µg rabbit IgG per condition for 1 h followed by incubation with 20 µl protein-G-agarose (Roche Applied Science, Penzberg, Germany) for 2 h. Precleared chromatin was incubated with H3K27me3 antibody (3 µg per condition) for 18 h at 4 °C. To collect immune complexes, 20 µl protein-G-agarose was added to the chromatin and incubated for additional 2 h at 4 °C. Unbound chromatin was kept as input for the subsequent sonication band check. Protein-G-agarose pellets were washed at 5000 × g; twice with 500 µl IP wash buffer 1 (20 mM Tris [pH 8.0], 2 mM EDTA 50 mM NaCl, 1% Triton-X100, and 0.1% SDS), once with IP wash buffer 2 (10 mM Tris [pH 8.0], 1 mM EDTA, 0.25 M LiCl, 1% NP-40, and 1% Sodium deoxycholate) and twice with TE (10 mM Tris, 1 mM EDTA [pH 8.0]). Immuno-precipitated chromatin was eluted in 300 µl Elution Buffer (100 mM NaHCO₃, 1% SDS) and cross-linking was reversed by incubation with RNase A and NaCl (0.3 M final concentration) at 67 °C for 18 h followed by treatment with Proteinase K at 45 °C for 2 h. Input DNA was treated with RNase A and Proteinase K simultaneously. DNA was extracted twice using phenol-chloroform followed by ethanol precipitation. Purified DNA was resuspended in 20 µl nuclease-free water, and sent on dry-ice for sequencing.

**ChIP sequencing**. Library preparation was performed by BGI (Hong Kong) using a variation of the Illumina's standard protocol. The workflow involves end repair of ChIP enriched DNA using T4 DNA polymerase, Klenow DNA polymerase and T4 polynucleotide kinase to generate blunt ended fragments. "A" bases were added to the 3′ ends using Klenow fragments (3′ to 5′ exo minus) to generate DNA fragments for ligation of adapters, which have a single "T" base overhang at their 3′ end. Adapters were ligated to the DNA fragments using DNA ligase. Adapter-modified DNA fragments were amplified by PCR (15 cycles) and size selected (200 ± 25 bp) by running PCR products on a 2% agarose gel and purifying using a QIAGEN Gel Extraction Kit (QIAGEN, #28704). The libraries were validated and sequenced using a HiSeq 2500 analyzer.

**Bioinformatics ChIP-Seq**. The raw sequencing reads were filtered for adapter contamination, low quality scores, and we also excluded reads in which more than 10% of bases were unknown. The filtered reads were aligned to the human genome (hg19) using the software package BWA[67] with standard parameters resulting in $20 \times 10^6$ reads for both, sh-LINC00173 and sh-control samples. Peak calling was performed using the software package MACS2 with options to detect broad peaks for histone modifications (--broad) and to filter sites with an adjusted $q$-value < 0.05 (--broad-cutoff 0.05). The MACS2[68] routine bdgdiff was used to identify differentially methylated regions. Annotations from the RefSeq database (http://www.ncbi.nlm.nih.gov/refseq/) were used to catalog the locations of proximal promoter (−1 kbp to TSS) and heatmaps were generated using the publicly available software seqMINER[69].

**UCSC tracks**. Histone-CHIP-Seq, DNA-methylation and RNA-Seq data for *LINC00173* from mature human blood cells were obtained through the Blueprint hub[24] from within the UCSC Genome Browser (http://genome.ucsc.edu/; Human Dec. 2013 (GRCh38/hg38) Assembly). CAGE-Seq tracks, Genomic Evolutionary Rate Profiling RS-Scores and RS-score $P$-values were downloaded from FANTOM CAT Browser[25] (http://fantom.gsc.riken.jp/cat/v1/#/genes/ENSG00000196668.3). Visualization was performed with the UCSC Genome browser.

**Statistics**. Statistical evaluations were carried out using two-sided Welsh's $t$-test accounting for unequal variances between two groups and one-way ANOVA with Dunnet's post hoc test for multiple comparisons for more than two groups. The level of significance was set at $P < 0.05$. All data are presented as mean ± s.d. or s.e.m. as indicated. Calculations were performed using GraphPad Prism 6 (STATCON) or R statistical language.

**Data availability**. ncRNA expression profiles, fingerprint ncRNAs and guilt by association results are publically available (www.lncScape.de). All raw data have been deposited in NCBI's Gene Expression Omnibus and are accessible through GEO Series accession numbers: GSE98633, GSE98697, GSE98791, GSE98830, GSE98829, GSE98854, GSE98946. All other remaining data are available within the Article and Supplementary Files, or available from the authors upon request.

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

## Acknowledgements

We thank Drs Z. Li and S. Meyer for critically reading the manuscript and helpful discussions, M. Ballmaier for cell sorting, K. Weber and B. Fehse for providing LeGO plasmids, and Dr. J. Pollack (Stanford Medical School) for performing RNA-seq. This work was supported by grants to J.H.K. from the German Research Foundation (DFG) within the Emmy Noether-Programme (KL-2374/2-1), and from the Madeleine Schick-edanz foundation. Ad.S. is supported by the Clinician-Scientist Program "Young Academy" of the Hannover Medical School. A.Sc. was supported by the DFG-Cluster of excellence REBIRTH. D.W., F.S., R.J. and M.N. were supported by the Hannover Bio-medical Research School. S.G. was supported by the Graduate School of the University of Ulm (funded by the Excellence Initiative of the German Federal and State Governments). F.K. and D.H. were supported by the Max-Eder program from the German Cancer Aid (#109420 and #111743). F.K. was also supported by a fellowship 2010/04 by the European Hematology Association; and by the DFG (SFB 1074, project A5). We acknowledge funding from Cancer Australia and the Anthony Rothe Foundation (D.B.), Cancer Institute of New South Wales (A.S. and J.W.) and the National Health and Medical Research Council (J.E.P.). D.B. is a Peter Doherty research fellow of the National Health and Medical Research Council of Australia. J.W. is a Future fellow of the Australian Research Council. This study makes use of data generated by the Blueprint Consortium. A full list of contributing investigators is available from www.blueprint-epigenome.eu. Funding for the project was provided by the European Union's Seventh Framework Programme (FP7/2007-2013) under grant agreement #282510 BLUEPRINT.

## Author contributions

S.E., F.S., M.N., F.F.A., D.W., S.K., C.R., R.J. and A.M. performed experiments and analyzed data. S.E. designed experiments and wrote the manuscript. Ad.S. performed bioinformatic analyses, designed and analyzed experiments and wrote the manuscript. M.Ng and A.Sc. revised the manuscript. F.F.A. programmed the online tool. D.H. designed experiments, analyzed data and revised the manuscript. M.J-L. sorted the human myelopoiesis samples. F.K., S.G., D.B., J.E.P., Y.H., J.T. and J.W.H.W. performed and analyzed ChIP and RNA sequencing. J.H.K. designed and supervised the study, analyzed data and wrote the manuscript. D.R. provided patient material and revised the manuscript.

## Additional information

**Competing interests:** The authors declare no competing financial interests.

