## [Peer Review File · Nature Communications]

Reviewers' comments:

Reviewer #2 (Remarks to the Author):

In this manuscript, the authors tried to define the non-coding RNA landscape of the human hematopoietic system. They identified fingerprint lncRNAs involved in blood homeostasis and uncovered prognostically relevant lncRNA stem cell signatures, suggesting the importance of the non-coding transcriptome for the formation and maintenance of the human blood hierarchy. However, the study is primary, mainly description; no novel mechanisms of lncRNAs involved in blood lineage or AML stem cell differentiation were provided.

Followings are my specific concerns:

1. The title of the manuscript is "Mapping the non-coding RNA landscape of human hematopoiesis and leukemia...", however, the study mainly focused on lncRNA, although a box C/D snoRNA and miRNAs cluster were mentioned, thus the title did not match the content of the manuscript.

2. LincRNA is one subtype of lncRNAs, however, the study used the term "lncRNAs and lincRNAs" and analyzed them separately, for example, page 6, lines 84-86 "lncRNAs and lincRNAs could be extracted for every blood cell population from the hotspots in the portraits (Fig. 2d and Supplementary Fig. 2d)". We question the accurate of the data analysis? An example is HOTAIRM1, which was also known as linc1548, a lincRNA identified previously and predicted to bind to PRC2 complex in the CLIP-seq data (doi:10.1038/nature10398), however, in this paper, HOTAIRM1 was considered as a lncRNA.

3. The RNA-FISH experiment of LINC00173 in fig. 4f-g might not be significant. The validation of LINC00173 binding to PRC2 is not enough only with RIP experiment. At least, the RNA-pull down experiment is necessary to show that EZH2 is one component of protein complex binding with LINC00173. Furthermore, it is not clear how LINC00173 represses the expression programs during granulopoiesis in fig. 4. Whether PRC2 which recruited by LINC00173 to participate in the granulopoiesis was not be uncovered yet in their work.

4. The authors have used three microarray platforms (lncRNA, miRNA, ncRNA), however the subsequent systematic analysis to distinguish blood cell populations in Figure 2 only focus on lncRNA and lincRNA. Whether miRNA data could be another ncRNA signature?

5. The number of fingerprint lncRNAs in the text (line 94, "2541 fingerprint") was not consistent with that shown in figure 2e. It should be "Fig. 5e" in line 237 and "7d-e" in line 301.

6. The description in the text from line 307 to line 309 is not consistent with the data shown in fig. 7b-c?

Reviewer #3 (Remarks to the Author):

This work investigates and uncovers a non-coding RNA expression system involved in the control of myeloid differentiation. The work processes from the generation of expression data, through complex and complete bioinformatic analysis, adds functional validation by silencing knockouts, and finally shows the lncRNA stem cell signature may be relevant in AML prognosis.

The paper is very readable, and the conclusions seem justified. The paper will go a long way in helping define the role of lncRNA in normal and abnormal hematopoiesis.

I do not believe further experiments are needed. I found the chromosome 14/megakaryopoiesis

story a bit distracting. Moreover in the Discussion, it might be of interest to compare the stem cell signature to other published mRNA stem cell signatures associated with AML. How much of the other mRNA signature does the authors lncRNA capture?

Reviewer #4 (Remarks to the Author):

The manuscript

Mapping the non-coding RNA landscape of human hematopoiesis and leukemia reveals stem cell non-coding gene expression programs

by S. Emmrich et al.

presents a comprehensive analysis of long non-coding RNA (lncRNA) expression portraits in 4 blood cell population, identified unique fingerprint lncRNAs and assigned these to critical regulatory circuits involved in blood. The paper highlights the importance of the non-coding transcriptome for the formation and maintenance of the human blood hierarchy homeostasis.

The study analyzed lncRNA abundance data and it also performed mRNA transcriptome analysis in parallel to assign the biological context of lncRNA expression based on the guilt-by-association principle.

A battery of bioinformatics methods is applied to the data, among them machine learning using self organizing maps (SOM). The authors use the oposSOM package to portray RNA landscapes of human hematopoietic cells. Normalization of the data and parametrization of the SOM algorithm was performed properly and agrees with default settings for unbiased data analysis. The results are sound and they are presented in an appealing way. Linkage of expression portraits and additional information like clustering tree and expression profiles leads to rounded characterization of the landscapes generated.

Under methodical aspects this application of SOM-portrayal reflects the power of the method to extract subtle differences between different cell types and to extract lists of marker genes which are then further filtered and verified by additional methods.

I have one point that the authors possibly want to address: SOM-portrayal characterizes the expression landscapes of each sample and as such this method is relatively sensitive for outliers and inconsistent expression patterns. The sample portraits shown in Supplementary Fig. 7 reveal partly a high degree of heterogeneity where cells of the same type partly show opposite expression patterns (e.g. MLL-AF9). Moreover, there seems to exist a bias between the first and the last samples in each row. Is there an explanation for this and how this heterogeneity affects the fingerprint markers and/or their interpretation in terms of function.

Reviewer #2

In this manuscript, the authors tried to define the non-coding RNA landscape of the human hematopoietic system. They identified fingerprint lncRNAs involved in blood homeostasis and uncovered prognostically relevant lncRNA stem cell signatures, suggesting the importance of the non-coding transcriptome for the formation and maintenance of the human blood hierarchy. However, the study is primary, mainly description; no novel mechanisms of lncRNAs involved in blood lineage or AML stem cell differentiation were provided.

REPOSE: We thank the reviewer for the thoughtful comments and constructive criticisms which certainly helped us to improve the manuscript. Our study provides a comprehensive non-coding RNA expression resource across the entire human hematopoietic system and AML. Such a resource – built from extremely difficult to obtain primary human material – has not been presented to date. Using three strong examples (*LINC00173*, *DLK1-DIO3* locus and prognostic lncRNA-stem cell signatures in AML) we further illustrate how our dataset and bioinformatic pipeline can be utilized to identify important regulatory circuits in human hematopoiesis and steer hypothesis-driven research. We believe that our work holds significant impact for the fields of hematology, oncology and stem cell biology, by enabling the investigation of novel lncRNA mechanisms. In the revised manuscript, we furthermore present additional experiments that provide new insights into how *LINC00173* regulates myelopoiesis.

Followings are my specific concerns:

1. The title of the manuscript is “Mapping the non-coding RNA landscape of human hematopoiesis and leukemia...”, however, the study mainly focused on lncRNA, although a box C/D snoRNA and miRNAs cluster were mentioned, thus the title did not match the content of the manuscript.

REPOSE: To provide a comprehensive non-coding RNA expression resource across the entire human hematopoietic system and AML, we used three different microarray platforms and complemented our data with RNA sequencing of the myeloid differentiation pathway. We quantified more than 20,000 GENCODE-annotated ncRNAs including scaRNAs, snoRNAs, snRNAs and all known miRNAs (Fig. 1b-c). Therefore, we feel that the title of our manuscript is justified. However, we agree with the reviewer that the previous manuscript mainly focused on lncRNAs. This was due to the large number of lncRNAs in the human genome, which far exceeds the number of small non-coding RNA genes. In the revised manuscript, we are more concise regarding the other classes of ncRNAs, and provide a more detailed breakdown of the transcripts covered in Fig. 1b-d.

2. LincRNA is one subtype of lncRNAs, however, the study used the term “lncRNAs and lincRNAs” and analyzed them separately, for example, page 6, lines 84-86 “lncRNAs and lincRNAs could be extracted for every blood cell population from the hotspots in the portraits (Fig. 2d and Supplementary Fig. 2d)”. We question the accurate of the data analysis? An example is HOTAIRM1, which was also known as linc1548, a lincRNA indentified previously and predicted to bind to PRC2 complex in the CLIP-seq data (doi:10.1038/nature10398), however, in this paper, HOTAIRM1 was considered as a lncRNA.

REPOSE: As the reviewer pointed out correctly, lincRNAs are a specific subclass of lncRNAs. They are transcribed from loci that do not overlap with protein coding genes, and can act in *trans* as scaffolds or guides for protein complexes. This makes lincRNAs attractive candidates for therapeutic intervention, which is why we decided to analyze them as a separate class, consistent with other ncRNA resources (Hon et al. Nature 2017). When we refer to lncRNAs, lincRNAs are also included. However, when referring to lincRNAs, other classes of lncRNAs or ncRNAs are excluded. In the revised manuscript, we clarify this to avoid confusion.

Throughout the manuscript we used GENCODEv23 (the reference human genome annotation for The ENCODE Project; released 07.2015) to annotate transcripts and discriminate between classes of non-coding RNAs. Within the scientific community, GENCODE is the most accepted and widely-used reference human genome annotation. GENCODEv23 classifies HOTAIRM1 (HOXA transcript antisense RNA, myeloid-specific 1 [HUGO]) as an antisense transcript within the HOXA locus, since some isoforms of HOTAIRM1 overlap with the first exon of HOXA1. The recently released MiTranscriptome (Iyer et al Nature Genetics 2015) shows additional overlaps between the 3' end of HOTAIRM1 and the 3' end of HOXA2. However, as pointed out by the reviewer, HOTAIRM1 indeed acts in *trans* like many lincRNAs, despite being now officially classified as an antisense-lincRNA.

3. The RNA-FISH experiment of *LINC00173* in fig. 4f-g might not be significant. The validation of *LINC00173* binding to PRC2 is not enough only with RIP experiment. At least, the RNA-pull down experiment is necessary to show that EZH2 is one component of protein complex binding with *LINC00173*. Furthermore, it is not clear how *LINC00173* represses the expression programs during granulopoiesis in fig. 4. Whether PRC2 which recruited by *LINC00173* to participate in the granulopoiesis was not be uncovered yet in their work.

RESPONSE: In the revised version of the manuscript, we have included novel data that provide further evidence for the proposed link between the granulocyte-specific non-coding RNA *LINC00173* and the PRC2 complex. In particular, we performed RNA-immunoprecipitation experiments with two antibodies for EZH2 (Fig. 4j) and another *LINC00173*-expressing cell line (Supplemental Fig. 5h) to prove the association of *LINC00173* with EZH2. In addition, we performed ChIP-seq experiments in primary human hematopoietic stem and progenitor cells to show that knockdown of *LINC00173* results in differential trimethylation of H3K27 at the loci of key stem cell genes (Fig. 4k).

RNA pulldown is technically challenging, and is an insensitive method that has only been reported for abundant lincRNAs – such as XIST – in cell lines (McHugh et al. Nature 2015 or Chu et al. Cell 2015). Therefore, we could not provide these data for a *LINC00173*, which is only abundantly expressed in primary mature neutrophils and lowly expressed in cell lines. However, we went a step further. Instead of only showing the binding of EZH2 and *LINC00173* with another method, we demonstrate molecular consequences of the loss of this interaction, i.e. perturbation of H3K27 trimethylation in *LINC00173* knockdown cells.

We moreover do not believe that including this method (RNA pulldown) would substantially add to the major aim of our study, which is to provide a comprehensive non-coding RNA expression resource across the entire human hematopoietic system and AML. *LINC00173* should serve as an example to illustrate how our dataset and bioinformatics pipeline can be utilized for hypothesis driven research.

4. The authors have used three microarray platforms (lincRNA, miRNA, ncRNA), however the subsequent systematic analysis to distinguish blood cell populations in Figure 2 only focus on lincRNA and lincRNA. Whether miRNA data could be another ncRNA signature?

RESPONSE: As correctly pointed out by the reviewer, blood cells can be distinguished by their unique miRNA profiles, which constitute additional ncRNA signatures. These miRNA signatures are provided in our resource (www.lincScape.de) and presented in Fig. 5a-b and the new Supplementary Fig. 6a of the revised manuscript: “As previously shown, each blood cell population possesses a distinct miRNA expression profile, which we confirmed with our dataset using *t*-SNE, LIMMA and SOM expression portraits (Fig. 5a-b and Supplementary Fig. 6a).” (page 12, paragraph 3).

Notably, although miRNA signatures have previously been published (Petriv et al. PNAS 2010; Arnold et al Genome Res 2011; Heiser et al Plos ONE 2014) and overlap with our findings, only the integration of

our comprehensive ncRNA resource uncovered a novel role of the *DLK1-DIO3*-locus in human megakaryopoiesis, which is why we devote a specific paragraph to this topic (page 12).

5. The number of fingerprint lncRNAs in the text (line 94, “2541 fingerprint”) was not consistent with that shown in figure 2e. It should be “Fig. 5e” in line 237 and “7d-e” in line 301.

REPOSE: We thank the reviewer for pointing out these mistakes, which we have corrected in the revised manuscript.

6. The description in the text from line 307 to line 309 is not consistent with the data shown in fig. 7b-c?

REPOSE: We thank the reviewer for pointing out this labeling mistake, which we have corrected in the revised manuscript.

Reviewer #3:

This work investigates and uncovers a non-coding RNA expression system involved in the control of myeloid differentiation. The work processes from the generation of expression data, through complex and complete bioinformatic analysis, adds functional validation by silencing knockouts, and finally shows the lncRNA stem cell signature may be relevant in AML prognosis.

The paper is very readable, and the conclusions seem justified. The paper will go a long way in helping define the role of lncRNA in normal and abnormal hematopoiesis.

I do not believe further experiments are needed. I found the chromosome 14/megakaryopoiesis story a bit distracting.

RESPONSE: We thank the reviewer for appreciating the complexity of the comprehensive nature our manuscript. Given its complexity, we understand the criticism of the reviewer regarding the *DLK1-DIO3*/chromosome 14 story. This part was included to illustrate how miRNA expression signatures could be used in conjunction with lncRNA expression and genomic data for hypothesis-driven research. In the revised manuscript, we improved upon the fit of *DLK1-DIO3*/chromosome 14 in the flow of the manuscript. For more consistent presentation of the data, we prepared the miRNA data in the same way as in Fig 2 (SOM + tree), which emphasizes that the miRNA signatures are an integral part of the resource.

Moreover in the Discussion, it might be of interest to compare the stem cell signature to other published mRNA stem cell signatures associated with AML. How much of the other mRNA signature does the authors lncRNA capture?

RESPONSE: We thank the reviewer for this suggestion. To address this, we performed a direct comparison of our SC-AML signature and the recently published 17 gene stemness score (LSC17; Ng et al. Nature 2017) as well as the earlier published version of the gene stemness signature (LSC.R; Eppert et al. Nat Medicine 2011). Both signatures (LSC17 and LSC.R) had been reported to indicate poor prognosis in adult AML patients. As published, a high LSC17 strongly predicted an adverse survival ($p < 0.0001$). Strikingly, we found that our lncRNA signatures could further stratify LSC17high and LSC17low patients and were independent risk predictors. These data are included as Supplemental Fig. 9 and mentioned on page 16, paragraph 2 of our revised manuscript. The data demonstrate how incorporation of ncRNAs will improve the prognostic value of published coding gene based signatures.

In contrast, the LSC.R score could not discern patients with poor prognosis in the validation cohort (TCGA-LAML; see Figure below). Therefore, we did not further consider this signature and did not include the data in the manuscript.

Reviewer #4:

Mapping the non-coding RNA landscape of human hematopoiesis and leukemia reveals stem cell non-coding gene expression programs by S. Emmrich et al. presents a comprehensive analysis of long non-coding RNA (lncRNA) expression portraits in 4 blood cell population, identified unique fingerprint lncRNAs and assigned these to critical regulatory circuits involved in blood. The paper highlights the importance of the non-coding transcriptome for the formation and maintenance of the human blood hierarchy homeostasis.

The study analyzed lncRNA abundance data and it also performed mRNA transcriptome analysis in parallel to assign the biological context of lncRNA expression based on the guilt-by-association principle.

A battery of bioinformatics methods is applied to the data, among them machine learning using self organizing maps (SOM). The authors use the oposSOM package to portray RNA landscapes of human hematopoietic cells. Normalization of the data and parametrization of the SOM algorithm was performed properly and agrees with default settings for unbiased data analysis. The results are sound and they are presented in an appealing way. Linkage of expression portraits and additional information like clustering tree and expression profiles leads to rounded characterization of the landscapes generated.

RESPONSE: We thank the reviewer for acknowledging the comprehensive nature of our manuscript.

Under methodical aspects this application of SOM-portrayal reflects the power of the method to extract subtle differences between different cell types and to extract lists of marker genes which are then further filtered and verified by additional methods.

I have one point that the authors possibly want to address: SOM-portrayal characterizes the expression landscapes of each sample and as such this method is relatively sensitive for outliers and inconsistent expression patterns. The sample portraits shown in Supplementary Fig. 7 reveal partly a high degree of heterogeneity where cells of the same type partly show opposite expression patterns (e.g. MLL-AF9). Moreover, there seems to exist a bias between the first and the last samples in each row. Is there an explanation for this and how this heterogeneity affects the fingerprint markers and/or their interpretation in terms of function.

RESPONSE: The reviewer raises an important issue. The stability of unsupervised clustering and dimensionality-reduction methods in high dimensional datasets are strongly dependent on the selection of input features. To assess stability of the clustering/SOM results and the conclusions drawn from it, all unsupervised analyses were performed several times using different hyperparameters (SOM size/Perplexity parameter in t-SNE) and different numbers of input features based on different cutoffs (ROSE or IQR, see Methods section). The results of two iterations using different cutoffs are shown below, demonstrating the stability of our analyses.

To rule out biases introduced by using mixed populations of cells, we FACS sorted all samples, including the AMLs. For the healthy blood samples this yielded pure cell populations with very homogenous SOM portraits, as presented in Supplementary Fig. 8 (formerly 7). However, as the reviewer pointed out correctly, the expression portraits for the AML samples are much more heterogeneous. As highly purified (FACS-sorted) blast populations were used for each sample, the portraits reflect the true heterogeneity of AML and cannot be attributed to contamination with other cells. This heterogeneity is best exemplified by the dissociation of the MLL-rearranged samples into two major groups as revealed by the SOMs and t-SNE (Fig. 6b and Supplementary Fig. 8b-c). The first group is comprised of immature MLLr-AMLs, which cluster near the HSCs. The other samples exhibit a more mature myeloid gene expression profile. This dichotomy in gene expression between MLLr-AML samples has been shown by several other groups (e.g. Valk et al. NEJM 2005) and might be due to different cells of origin (Stavropoulou et al. Cancer Cell 2016). To overcome these obstacles and to determine a high confidence MLL-ncRNA

signature, we selected samples with a high expression of MLLr-associated coding genes as described in the Methods section. Taken together, using these strategies, we could derive ncRNA signatures – shared between normal HSCs and AML blasts – that were confirmed using the independent TCGA dataset. We could show that both signatures can cluster the AML patients into groups with a different prognosis and outcomes (Fig. 7).

Stability of t-SNE and SOM analysis in healthy hematopoiesis

Stability of t-SNE and SOM analysis AML landscape

MLL-*AF9* 7094 ncRNAs as in Figure S8

MLL-*AF9* 5688 ncRNAs

REVIEWERS' COMMENTS:

Reviewer #2 (Remarks to the Author):

The authors have addressed the concerns we have raised. The subject matter is of a high level of interest in human hematopoiesis and AML. The bioinformatic analysis and datasets may steer more hypothesis-driven research of the noncoding RNA functions in human blood hierarchy homeostasis. Therefore, this manuscript is suitable for publication in Nature Communications.

Reviewer #3 (Remarks to the Author):

The authors have satisfied my questions and concerns. I thank them for their diligence.

Reviewer #4 (Remarks to the Author):

no further comments